# Fluctuations of finite-time Lyapunov exponents in an intermediate-complexity atmospheric model: a multivariate and large-deviation perspective

Frank Kwasniok

Department of Mathematics, University of Exeter, Exeter, United Kingdom

**Correspondence:** Frank Kwasniok (F.Kwasniok@exeter.ac.uk)

**Abstract.** The stability properties as characterised by the fluctuations of finite-time Lyapunov exponents around their mean values are investigated in a three-level quasi-geostrophic atmospheric model with realistic mean state and variability. Firstly, the covariance structure of the fluctuation field is examined. In order to identify dominant patterns of collective excitation an empirical orthogonal function (EOF) analysis of the fluctuation field of all of the finite-time Lyapunov exponents is performed. The three leading modes are patterns where the most unstable Lyapunov exponents fluctuate in phase. These modes are virtually independent of the integration time of the finite-time Lyapunov exponents. Secondly, large-deviation rate functions are estimated from time series of finite-time Lyapunov exponents based on the probability density functions and using the Legendre transform method. Serial correlation in the time series is properly accounted for. A large-deviation principle can be established for all of the Lyapunov exponents. Convergence is rather slow for the most unstable exponent, becomes faster when going further down in the Lyapunov spectrum, is very fast for the near-neutral and weakly dissipative modes, and turns slow again for the strongly dissipative modes at the end of the Lyapunov spectrum. The curvature of the rate functions at the minimum is linked to the corresponding elements of the diffusion matrix. Also the joint large-deviation rate function for the first and the second Lyapunov exponent is estimated.

## 1 Introduction

The atmosphere is a high-dimensional nonlinear chaotic dynamical system; its time evolution is characterised by sensitivity to initial conditions (Lorenz, 1963; Kalnay, 2003). As a consequence predictability is limited; small errors in the initial states progressively grow under the time evolution until the forecast eventually becomes useless, that is, indistinguishable from the invariant measure or climatology of the system. Understanding the structure of this inherent instability is key to improve forecasts at all timescales.

Sensitivity to initial conditions and perturbation growth in nonlinear dynamical systems are often quantified using Lyapunov exponents (LEs) (e.g., Eckmann and Ruelle, 1985; Ott, 2002; Pikovsky and Politi, 2016). They describe the asymptotic growth or decay of infinitesimal perturbations. A system is chaotic if it has at least one positive Lyapunov exponent. However, the predictability properties may vary substantially across state space. Finite-time (or local) Lyapunov exponents (FTLEs) allow a characterisation of the stability of a particular initial state with respect to a predefined prediction horizon.

LEs have been calculated for various geophysical fluid systems, ranging from highly truncated atmospheric models (Legras and Ghil, 1985), to intermediate-complexity atmospheric models (Vannitsem and Nicolis, 1997; Schubert and Lucarini, 2015), to coupled atmosphere-ocean models (Vannitsem and Lucarini, 2016). A review has been published recently by Vannitsem (2017). Models tuned to realistic conditions turn out to possess quite a large number of positive LEs corresponding to a high-dimensional chaotic attractor.

The present paper investigates the fluctuations of FTLEs in an intermediate-complexity atmospheric model with realistic mean state and variability. It focuses on two aspects which have so far found little attention in the context of geophysical fluid systems. Firstly, the covariance structure of the fluctuation field of the FTLEs is studied by means of a principal component (PC) or empirical orthogonal function (EOF) analysis (Kuptsov and Politi, 2011). Secondly, we are looking at the large-deviation behaviour of the FTLEs at long integration times (Kuptsov and Politi, 2011; Laffargue et al., 2013; Johnson and Meneveau, 2015). A large-deviation principle links the FTLEs at long integration times to the global LEs by providing a universal law for the probability density of fluctuations of the FTLEs around the mean value. It can be expected to hold for Axiom A dynamical systems and, invoking the chaotic hypothesis, also for certain types of non-Axiom A systems. In particular, a large-deviation law allows to determine the probability of very stable or very unstable atmospheric states.

The paper is organised as follows: In section 2 the atmospheric model is described. The methodology which consists of calculating LEs, the multivariate fluctuation analysis and the large-deviation theory is outlined in sections 3, 4 and 5. The results are presented and discussed in section 6. Some conclusions are drawn in section 7.

## 2   The atmospheric model

A quasi-geostrophic (QG) three-level model on the sphere, formulated in pressure coordinates, is used here as dynamical framework. The model is identical to that introduced by Kwasniok (2007) except for the horizontal resolution and the coefficient of hyperviscosity. A very similar model was introduced by Marshall and Molteni (1993). The dynamical equations are

$$\frac{\partial q_i}{\partial t} + J(\Psi_i, q_i) = D_i + S_i, \qquad i = 1, 2, 3 \tag{1}$$

where $\Psi_i$ and $q_i$ are the streamfunction and the potential vorticity at level $i$ and $J$ denotes the Jacobian operator on the sphere. All variables are nondimensional using the radius of the earth as the unit of length and the inverse of the angular velocity of the earth as the unit of time. The three pressure levels are located at 250, 500 and 750 hPa. Potential vorticity and the streamfunction are related by

$$q_1 = \nabla^2 \Psi_1 - R_{1,2}^{-2}(\Psi_1 - \Psi_2) + f \tag{2}$$

$$q_2 = \nabla^2 \Psi_2 + R_{1,2}^{-2}(\Psi_1 - \Psi_2) - R_{2,3}^{-2}(\Psi_2 - \Psi_3) + f \tag{3}$$

$$q_3 = \nabla^2 \Psi_3 + R_{2,3}^{-2}(\Psi_2 - \Psi_3) + f + f_0 h \tag{4}$$

where $\nabla$ is the horizontal gradient operator and $f$ is the Coriolis parameter. The Rossby deformation radii $R_{1,2}$ and $R_{2,3}$ have dimensional values of 575 km and 375 km. The function $h = h(\lambda, \mu)$ represents a nondimensional topography which is related

to the actual dimensional topography of the earth $h^* = h^*(\lambda, \mu)$ by $h = h^*/H$, where $H$ is a scale height set to 8 km and $f_0$ is the Coriolis parameter at an average geographic latitude taken to be $45°$N.

The dissipative terms are given as

$$D_1 = \tau_N^{-1} R_{1,2}^{-2}(\Psi_1 - \Psi_2) - k_H \nabla^8 \hat{q}_1 \tag{5}$$

$$D_2 = -\tau_N^{-1} R_{1,2}^{-2}(\Psi_1 - \Psi_2) + \tau_N^{-1} R_{2,3}^{-2}(\Psi_2 - \Psi_3) - k_H \nabla^8 \hat{q}_2 \tag{6}$$

$$D_3 = -\tau_N^{-1} R_{2,3}^{-2}(\Psi_2 - \Psi_3) - \tau_E^{-1} \nabla^2 \Psi_3 - k_H \nabla^8 \hat{q}_3 \tag{7}$$

They are Newtonian temperature relaxation with a radiative timescale of $\tau_N = 25$ days, Ekman damping on the lowest level with a spindown timescale of $\tau_E = 1.5$ days, and a strongly scale-selective horizontal diffusion of vorticity and temperature. The $\hat{q}_i$ is the time-dependent part of the potential vorticity at level $i$, that is, $\hat{q}_i = q_i - f - \delta_{i3} f_0 h$. The coefficient of horizontal diffusion $k_H = \tau_H^{-1}[n_m(n_m + 1)]^{-4}$ is such that harmonics of total wavenumber $n_m = 21$ are damped at a timescale of $\tau_H = 1.5$ days. The terms $S_i = S_i(\lambda, \mu)$ are diabatic sources of potential vorticity which are independent of time but spatially varying.

The model is considered on the northern hemisphere. The boundary condition of no meridional flow, $v_i(\lambda, 0) = 0$, that is, vanishing streamfunction, $\Psi_i(\lambda, 0) = 0$, is applied at the equator on all three model levels. The horizontal discretization is spectral, triangularly truncated at total wavenumber $n_m = 21$. The number of degrees of freedom is 231 for each level and $N = 693$ in total. The model is integrated in time using the third-order Adams–Bashforth scheme with a constant step size of 1 h.

The variables of the QG model are listed in Table 1; the model parameters are listed in Table 2 with their dimensional and nondimensional values.

In order to get a model behaviour close to that of the real atmosphere, the forcing terms $S_i$ are determined from ECMWF reanalysis data by requiring that when computing potential vorticity tendencies for a large number of observed atmospheric fields, the average of these tendencies must be zero (Roads, 1987), in order for the ensemble of reanalysis data states to be representative of a statistically stable long-term behaviour of the QG model. The timescale of horizontal diffusion $\tau_H$ is determined such that the slope of the kinetic energy spectrum at the truncation level in the model matches that in the reanalysis data. See Kwasniok (2007) for details on the parameter tuning procedure. The QG model exhibits in a long-term integration a remarkably realistic mean state and variability pattern of streamfunction and potential vorticity (see Table 3).

## 3  Lyapunov exponents

We consider a nonlinear autonomous dynamical system with state vector $\mathbf{x} = (x_1, \dots, x_N)^T$ governed by the evolution equations

$$\frac{d\mathbf{x}}{dt} = \mathbf{f}(\mathbf{x}). \tag{8}$$

The linearised dynamics of a small perturbation $\delta\mathbf{x}$ are given as

$$\frac{d}{dt}\delta\mathbf{x} = \frac{\partial \mathbf{f}}{\partial \mathbf{x}}\delta\mathbf{x}. \tag{9}$$

**Table 1.** Variables and fields in the QG model and their nondimensionalization with the earth radius $a = 6.371 \times 10^6 \, \mathrm{m}$ and the angular velocity of the earth $\Omega = 7.292 \times 10^{-5} \, \mathrm{s}^{-1}$.

| Symbol | | Description | Unit | Nondimensionalisation |
|---|---|---|---|---|
| $t$ | | time | s | $\Omega^{-1}$ |
| $\lambda$ | $(0 \leq \lambda < 2\pi)$ | longitude (eastward) | | |
| $\phi$ | $(0 \leq \phi \leq \pi/2)$ | latitude | | |
| $\mu = \sin\phi$ | $(0 \leq \mu \leq 1)$ | sine of latitude | | |
| $\Psi_i$ | | streamfunction at level $i$ | $\mathrm{m}^2\,\mathrm{s}^{-1}$ | $a^2\Omega$ |
| $q_i$ | | potential vorticity at level $i$ | $\mathrm{s}^{-1}$ | $\Omega$ |
| $\hat{q}_i$ | | time-dependent part of potential vorticity at level $i$ | $\mathrm{s}^{-1}$ | $\Omega$ |
| $f = 2\mu$ | | Coriolis parameter | $\mathrm{s}^{-1}$ | $\Omega$ |
| $h$ | | topography of the earth | m | $H$ |
| $S_i$ | | diabatic forcing at level $i$ | $\mathrm{s}^{-2}$ | $\Omega^2$ |
| $u_i = -\sqrt{1-\mu^2}\,(\partial\Psi_i/\partial\mu)$ | | zonal velocity (eastward) | $\mathrm{m}\,\mathrm{s}^{-1}$ | $a\Omega$ |
| $v_i = (1/\sqrt{1-\mu^2})\,(\partial\Psi_i/\partial\lambda)$ | | meridional velocity (northward) | $\mathrm{m}\,\mathrm{s}^{-1}$ | $a\Omega$ |

**Table 2.** Parameters in the QG model

| Symbol | Description | Dimensional value | Nondimensional value |
|---|---|---|---|
| $R_{1,2}$ | Rossby deformation radius between levels 1 and 2 | 575 km | $9.025 \times 10^{-2}$ |
| $R_{2,3}$ | Rossby deformation radius between levels 2 and 3 | 375 km | $5.886 \times 10^{-2}$ |
| $\tau_\mathrm{N}$ | timescale of temperature relaxation | 25 days | $50\pi$ |
| $\tau_\mathrm{E}$ | timescale of Ekman damping | 1.5 days | $3\pi$ |
| $\tau_\mathrm{H}$ | timescale of horizontal diffusion at wavenumber 21 | 1.5 days | $3\pi$ |
| $f_0$ | Coriolis parameter at 45°N | $1.031 \times 10^{-4}\,\mathrm{s}^{-1}$ | $\sqrt{2}$ |
| $H$ | scale height | 8 km | |

The propagation of the perturbation between time $t_0$ with initial state $\mathbf{x}_0 = \mathbf{x}(t_0)$ and time $t$ $(t > t_0)$ can be written as

$$\delta\mathbf{x}(t) = \mathbf{M}(\mathbf{x}_0, t - t_0)\delta\mathbf{x}(t_0) \tag{10}$$

where $\mathbf{M}$ is the resolvent matrix. If the system is ergodic then according to the theorem by Oseledets (1968) the limit

$$\mathbf{S} = \lim_{t\to\infty} (\mathbf{M}^\mathrm{T}\mathbf{M})^{\frac{1}{2(t-t_0)}} \tag{11}$$

5  exists and is the same for almost all initial conditions $\mathbf{x}_0$. The (global) LEs are defined as

$$\lambda_j = \log\omega_j, \qquad j = 1, \ldots, N, \tag{12}$$

**Table 3.** Pattern correlation of various fields in the QG model with the corresponding fields in ECMWF reanalysis data

| Level | $\langle \Psi_i \rangle$ | $\sqrt{\langle \Psi_i'^2 \rangle}$ | $\sqrt{\langle \hat{q}_i'^2 \rangle}$ |
|---|---|---|---|
| 250 hPa | 0.99 | 0.99 | 0.97 |
| 500 hPa | 0.99 | 0.99 | 0.98 |
| 750 hPa | 0.96 | 0.97 | 0.94 |

where $\{\omega_j\}_{j=1}^N$ are the positive eigenvalues of the matrix $\mathbf{S}$. The set of all LEs, usually presented in non-increasing order, is called the *Lyapunov spectrum*. The LEs are independent of norm.

In order to characterise perturbation growth or decay over a finite integration time $\tau$ the FTLEs $\Lambda_j^{(\tau)}(\mathbf{x}_0)$ are introduced. There are three different definitions of FTLEs. One can compute them by making reference to the backward, forward or covariant Lyapunov vectors; see, e.g., Kuptsov and Parlitz, 2012 for a review. In the limit of large integration time $\tau$, which is the main focus of the present study, all of the three definitions become more and more equivalent (Kuptsov and Politi, 2011; Pazó et al., 2013). We here refer to the backward FTLEs as they are easiest to compute. They are calculated using the standard algorithm based on the Gram–Schmidt orthogonalisation (Shimada and Nagashima, 1979; Benettin et al., 1980). An ensemble of $N$ linearly independent perturbations is initialised and integrated forward in time together with the nonlinear model trajectory. A transient period is discarded for the trajectory to settle on the attractor of the system and the perturbations to converge to the backward Lyapunov vectors. Then after every integration time interval $\Delta\tau$ the perturbations are re-orthonormalised using a QR-decomposition performed via the Gram–Schmidt procedure. The FTLEs are obtained as

$$\Lambda_j^{(\Delta\tau)}(\mathbf{x}_\alpha) = \Lambda_{j,\alpha}^{(\Delta\tau)} = \frac{1}{\Delta\tau} \log R_{jj}(t_\alpha, t_{\alpha+1}), \qquad \alpha = 0, \ldots, L-1, \tag{13}$$

where $R_{jj}(t_\alpha, t_{\alpha+1})$ are the diagonal elements of the triangular matrix $\mathbf{R}$ in the QR-decomposition resulting from the integration between times $t_\alpha$ and $t_{\alpha+1}$. We have $t_\alpha = t_0 + \alpha\Delta\tau$ and $\mathbf{x}_\alpha = \mathbf{x}(t_\alpha)$. The FTLEs $\Lambda_j^{(\tau)}$ for larger integration times $\tau = n\Delta\tau$ are obtained by averaging over $n$ consecutive values of $\Lambda_j^{(\Delta\tau)}$:

$$\Lambda_{j,\alpha}^{(\tau)} = \frac{1}{n} \sum_{i=0}^{n-1} \Lambda_{j,\alpha+i}^{(\Delta\tau)}, \qquad \alpha = 0, \ldots, L-1. \tag{14}$$

For all integration times $\tau$, we keep time series of FTLEs of the same length $L$, $\{\Lambda_{j,\alpha}^{(\tau)}\}_{\alpha=0}^{L-1}$, characterising the stability of the states $\{\mathbf{x}_\alpha\}_{\alpha=0}^{L-1}$ over the time horizon $\tau$.

The FTLEs depend on the scalar product chosen in the Gram–Schmidt orthogonalisation procedure. We here use the total energy scalar product with its associated total energy norm (Ehrendorfer, 2000; Kwasniok, 2007). The dependence of the FTLEs on the norm becomes increasingly weaker in the limit of large integration time $\tau$.

The FTLEs are related to the global LEs by

$$\lim_{\tau \to \infty} \Lambda_j^{(\tau)}(\mathbf{x}_0) = \lambda_j \tag{15}$$

for almost all initial states $\mathbf{x}_0$ and

$$\left\langle \Lambda_j^{(\tau)} \right\rangle = \lambda_j \tag{16}$$

for all $\tau$ where $\langle \cdot \rangle$ denotes an ensemble average over the attractor of the system which for ergodic systems can be estimated as a mean over a long time series.

## 4 Multivariate fluctuation analysis

The vector of global Lyapunov exponents is defined as $\boldsymbol{\lambda} = (\lambda_1, \ldots, \lambda_N)^{\mathrm{T}}$ and the fluctuation field as $\boldsymbol{\Lambda}^{(\tau)} - \boldsymbol{\lambda} = \left( \Lambda_1^{(\tau)} - \lambda_1, \ldots, \Lambda_N^{(\tau)} - \lambda_N \right)^{\mathrm{T}}$. We study the correlations between the fluctuations of the FTLEs; to do this, preferred patterns of collective excitation are extracted. A canonical approach is a principal component (PC) or empirical orthogonal function (EOF) analysis based on the scaled covariance matrix $\mathbf{D}^{(\tau)}$ defined as

$$\mathbf{D}^{(\tau)} = \left\langle \left( \boldsymbol{\Lambda}^{(\tau)} - \boldsymbol{\lambda} \right) \left( \boldsymbol{\Lambda}^{(\tau)} - \boldsymbol{\lambda} \right)^{\mathrm{T}} \right\rangle \tau = \frac{\tau}{L} \sum_{\alpha=0}^{L-1} \left( \boldsymbol{\Lambda}_\alpha^{(\tau)} - \boldsymbol{\lambda} \right) \left( \boldsymbol{\Lambda}_\alpha^{(\tau)} - \boldsymbol{\lambda} \right)^{\mathrm{T}}. \tag{17}$$

In the limit of large integration time $\tau$ we expect convergence to the diffusion matrix $\mathbf{D}$ (Kuptsov and Politi, 2011; Pikovsky and Politi, 2016):

$$\mathbf{D} = \lim_{\tau \to \infty} \mathbf{D}^{(\tau)} \tag{18}$$

The eigenvalues and eigenvectors of the symmetric, positive definite matrix $\mathbf{D}^{(\tau)}$ are calculated:

$$\mathbf{D}^{(\tau)} \mathbf{e}_j^{(\tau)} = \nu_j^{(\tau)} \mathbf{e}_j^{(\tau)} \tag{19}$$

The eigenvalues $\{\nu_j^{(\tau)}\}_{j=1}^{N}$ are arranged in non-increasing order. The eigenvectors form an orthonormal system:

$$\mathbf{e}_j^{(\tau)} \cdot \mathbf{e}_k^{(\tau)} = \delta_{jk} \tag{20}$$

The fluctuation field of the FTLEs is expanded as

$$\boldsymbol{\Lambda}_\alpha^{(\tau)} - \boldsymbol{\lambda} = \sum_{j=1}^{N} y_{j,\alpha}^{(\tau)} \mathbf{e}_j^{(\tau)} \tag{21}$$

with $y_{j,\alpha}^{(\tau)} = \mathbf{e}_j^{(\tau)} \cdot (\boldsymbol{\Lambda}_\alpha^{(\tau)} - \boldsymbol{\lambda})$. The principal components $\{y_j^{(\tau)}\}_{j=1}^{N}$ are uncorrelated and their variance is given by the corresponding eigenvalue:

$$\left\langle y_j^{(\tau)} y_k^{(\tau)} \right\rangle = \frac{1}{L} \sum_{\alpha=0}^{L-1} y_{j,\alpha}^{(\tau)} y_{k,\alpha}^{(\tau)} = \nu_j^{(\tau)} \delta_{jk} \tag{22}$$

The steepness or complexity of the eigenvalue spectrum is characterised by the fraction of variance explained by the principal component $y_j^{(\tau)}$ given as

$$r_j^{(\tau)} = \frac{\nu_j^{(\tau)}}{\sum_{k=1}^{N} \nu_k^{(\tau)}} \tag{23}$$

and the cumulative fraction of variance given as

$$c_j^{(\tau)} = \frac{\sum_{k=1}^{j} \nu_k^{(\tau)}}{\sum_{k=1}^{N} \nu_k^{(\tau)}} \tag{24}$$

As a possible further step, one may try to link the covariance structure of the FTLEs with investigations of the angles between the covariant Lyapunov vectors and the degree of entanglement and interaction of the various unstable and stable directions in tangent space (Yang et al., 2009). This is related to the hyperbolicity and the inertial manifold of the system.

## 5 Large-deviation theory for FTLEs

Large-deviation theory (Kifer, 1990; Touchette, 2009) is a powerful approach from statistical physics for estimating the probability of rare events with many applications. It has recently been applied to the behaviour of FTLEs at long integration times (Kuptsov and Politi, 2011; Laffargue et al., 2013; Johnson and Meneveau, 2015). Large-deviation theory is in the following briefly described in the form in which it is used in the present study.

### 5.1 Univariate theory

For a sequence of $n$ identically distributed but not necessarily independent random variables, $\{X_i\}_{i=1}^{n}$, the sample mean

$$A_n = \frac{1}{n} \sum_{i=1}^{n} X_i \tag{25}$$

is an unbiased estimator of and converges to the true mean, $\langle X \rangle$, as $n \to \infty$. By the Gärtner–Ellis theorem (Touchette, 2009), if the *scaled cumulant generating function* (SCGF)

$$\gamma(\theta) = \lim_{n\to\infty} \frac{1}{n} \log \langle e^{n\theta A_n} \rangle \tag{26}$$

exists and is differentiable everywhere then $A_n$ follows a large-deviation principle,

$$p(A_n = z) \sim \exp[-nI(z)], \tag{27}$$

where the *large-deviation rate function* $I(z)$ is independent of $n$ and given as the *Legendre–Fenchel transform* of the SCGF:

$$I(z) = \sup_{\theta \in \mathbb{R}} [\theta z - \gamma(\theta)] \tag{28}$$

The rate function $I(z)$ is non-negative and strictly convex. It has a unique zero and minimum at $z^* = \langle X \rangle$, that is, $I(\langle X \rangle) = 0$ and $I'(\langle X \rangle) = 0$. The curvature of the rate function at the minimum is given as (Touchette, 2009)

$$I''(\langle X \rangle) = \frac{1}{\lim_{n\to\infty} n \langle (A_n - \langle A_n \rangle)^2 \rangle}. \tag{29}$$

In view of eq.(14), FTLEs immediately lend themselves to large-deviation theory. For large integration time $\tau$, one would expect the probability density of the FTLE $\Lambda_j^{(\tau)}$ to follow a large-deviation principle,

$$p\left(\Lambda_j^{(\tau)} = z\right) \sim \exp[-\tau I_j(z)], \tag{30}$$

where the large-deviation rate function $I_j(z)$ ~~which~~ is independent of $\tau$ and given as

$$I_j(z) = \sup_{\theta \in \mathbb{R}}[\theta z - \gamma_j(\theta)] \tag{31}$$

with the SCGF

$$\gamma_j(\theta) = \lim_{\tau \to \infty} \frac{1}{\tau} \log \left\langle e^{\tau \theta \Lambda_j^{(\tau)}} \right\rangle. \tag{32}$$

Introducing $\theta' = \tau \theta$ and then dropping the prime again we get

$$I_j(z) = \lim_{\tau \to \infty} \frac{1}{\tau} \sup_{\theta \in \mathbb{R}} \left[ \theta z - \log \left\langle e^{\theta \Lambda_j^{(\tau)}} \right\rangle \right]. \tag{33}$$

We expect convergence of the rate function $I_j(z)$ as soon as the integration time $\tau$ is large enough for consecutive values of $\Lambda_j^{(\tau)}$ taken over non-overlapping integration time intervals, $\Lambda_{j,\alpha}^{(\tau)}$ and $\Lambda_{j,\alpha+n}^{(\tau)}$, to be independent. This is actually an application of the *block averaging method* (Rohwer et al., 2015). Note, however, that convergence of the rate function at a particular value of $\tau$ here does not guarantee that the probability density function is already in the large-deviation limit at that value of $\tau$.

The rate function $I_j(z)$ has a unique zero and minimum at $z^* = \lambda_j$, that is, $I_j(\lambda_j) = 0$ and $I_j'(\lambda_j) = 0$. The curvature of the rate function at the minimum is linked to the diffusion matrix $\mathbf{D}$ as

$$I_j''(\lambda_j) = \frac{1}{\lim_{\tau \to \infty} \left\langle \left(\Lambda_j^{(\tau)} - \lambda_j\right)^2 \right\rangle \tau} = D_{j,j}^{-1} \tag{34}$$

A second-order Taylor expansion of the rate function in the vicinity of $\lambda_j$,

$$I_j(z) \approx \frac{1}{2} I_j''(\lambda_j)(z - \lambda_j)^2, \tag{35}$$

corresponds to a Gaussian probability density with mean $\lambda_j$ and variance $D_{j,j}/\tau$, recovering the central limit theorem (CLT) as a limit case of large-deviation theory.

## 5.2 Estimating the rate function

There are two ways of estimating the rate functions $I_j(z)$ from data: via the probability density function (cf., eq.(30)) or via the Legendre transform (cf., eq.(33)).

### 5.2.1 Probability density function approach

By inverting eq.(30) we have

$$I_j(z) = - \lim_{\tau \to \infty} \frac{1}{\tau} \log p\left(\Lambda_j^{(\tau)} = z\right). \tag{36}$$

We take a maximum likelihood approach for estimating the rate function. The probability density of $\Lambda_j^{(\tau)}$ is modelled as

$$p\left(\Lambda_j^{(\tau)} = z\right) = \frac{1}{Z_j^{(\tau)}} \exp\left[-U_j^{(\tau)}(z)\right] \tag{37}$$

with normalisation constant

$$Z_j^{(\tau)} = \int\limits_{-\infty}^{\infty} \exp\left[-U_j^{(\tau)}(z)\right] dz. \tag{38}$$

The potential function $U_j^{(\tau)}(z)$ is expanded into a polynomial basis in standardized variables:

$$U_j^{(\tau)}(z) = \sum_{i=1}^{M} \beta_i^{(\tau)} \left(\frac{z - \lambda_j}{\sigma_j^{(\tau)}}\right)^i \tag{39}$$

Here $\sigma_j^{(\tau)}$ is the standard deviation of the FTLE $\Lambda_j^{(\tau)}$:

$$\sigma_j^{(\tau)} = \left\langle \left(\Lambda_j^{(\tau)} - \lambda_j\right)^2 \right\rangle^{1/2} = \left[\frac{1}{L} \sum_{\alpha=0}^{L-1} \left(\Lambda_{j,\alpha}^{(\tau)} - \lambda_j\right)^2\right]^{1/2} \tag{40}$$

The parameter $M$ determines the complexity of the model. In order to have a normalisable probability density, we need $M$
to be even and $\beta_M^{(\tau)} > 0$. The expansion coefficients $\{\beta_i^{(\tau)}\}_{i=1}^{M}$ are determined by maximising the likelihood function of the data $\{\Lambda_{j,\alpha}^{(\tau)}\}_{\alpha=0}^{L-1}$. This is a convex optimisation problem with a unique maximum which is numerically stable to solve. Model selection is performed with the Bayesian information criterion.

The estimate of the rate function is given as

$$I_j(z) = \lim_{\tau \to \infty} \frac{1}{\tau} \left[U_j^{(\tau)}(z) - U_j^{(\tau)}(z^*)\right] \tag{41}$$

where $z^*$ denotes the position of the minimum of the potential function $U_j^{(\tau)}(z)$. Note that, for finite $\tau$, we do not necessarily have $z^* = \lambda_j$ as the mode of the probability density of $\Lambda_j^{(\tau)}$ may be different from its mean if the distribution is skewed; but we always have $z^* \to \lambda_j$ as $\tau \to \infty$. One would now estimate $I_j(z)$ from the probability density function of $\Lambda_j^{(\tau)}$ for various large values of $\tau$ and look for convergence.

The maximum likelihood method tends to provide very smooth and convex rate functions although convexity is not strictly
guaranteed. It clearly improves on earlier work (e.g., Johnson and Meneveau, 2015) using histogram or kernel density estimates for the probability density and treating the normalisation constant only in the Gaussian approximation.

### 5.2.2   Legendre transform approach

Alternatively, the rate functions $I_j(z)$ can be determined by numerically implementing the Legendre transform of eq.(33) (Rohwer et al., 2015) with the moment generating function estimated by the sample mean over the time series:

$$\left\langle e^{\theta \Lambda_j^{(\tau)}} \right\rangle = \frac{1}{L} \sum_{\alpha=0}^{L-1} e^{\theta \Lambda_{j,\alpha}^{(\tau)}} \tag{42}$$

For each $z$, this is a convex optimisation problem with a unique solution if any exists. Rate functions obtained via the Legendre transform method are guaranteed to be strictly convex with a unique zero and minimum at $z^* = \lambda_j$.

Rate function estimates from the Legendre transform method converge as soon as $\tau = n\Delta\tau$ is large enough for successive values of $\Lambda_j^{(\tau)}$ over non-overlapping integration time intervals, $\Lambda_{j,\alpha}^{(\tau)}$ and $\Lambda_{j,\alpha+n}^{(\tau)}$, to be independent. However, this gives no indication whether or not the probability density function actually is already in the large-deviation limit. Therefore we here consider both rate function estimates side by side.

### 5.3 Estimating the diffusion coefficients

The diffusion coefficients $D_{j,j}$ can be obtained from both rate function estimates as the inverse of the curvature at the minimum (cf., eq.(34). They can also be estimated directly from the time series of the FTLEs according to eq.(17). It can be shown that the estimates from the Legendre transform-based rate function and from the time series are always the same; any differences just stem from the error of the finite-difference approximation of the curvature as the Legendre transform is not available in closed form. For a Gaussian probability density model, that is, $M = 2$ in eq.(39), the diffusion coefficient estimates from the probability density-based rate function and from the time series are exactly the same; otherwise they are different.

### 5.4 Multivariate theory

The large-deviation analysis can be extended to a multivariate approach (Kuptsov and Politi, 2011; Johnson and Meneveau, 2015). Let now $\mathbf{\Lambda}^{(\tau)}$ denote the column vector of any $K$-dimensional subset of the $N$ FTLEs and $\boldsymbol{\lambda}$ the corresponding vector of global LEs. We have $1 \leq K \leq N$ where $K = N$ corresponds to the full system and $K = 1$ recovers the univariate analysis. For large integration time $\tau$, the joint probability density function of the $K$ FTLEs would then follow a large-deviation principle,

$$p\left(\mathbf{\Lambda}^{(\tau)} = \mathbf{z}\right) \sim \exp[-\tau I(\mathbf{z})], \tag{43}$$

where the *joint large-deviation rate function $I(\mathbf{z})$* is independent of $\tau$ and given as the *multivariate Legendre–Fenchel transform*

$$I(\mathbf{z}) = \lim_{\tau \to \infty} \frac{1}{\tau} \sup_{\boldsymbol{\theta} \in \mathbb{R}^K} \left[\boldsymbol{\theta}^{\mathrm{T}}\mathbf{z} - \log\left\langle e^{\boldsymbol{\theta}^{\mathrm{T}}\mathbf{\Lambda}^{(\tau)}}\right\rangle\right] \tag{44}$$

The joint rate function $I(\mathbf{z})$ is non-negative and strictly convex. It has a unique zero and minimum at $\mathbf{z}^* = \boldsymbol{\lambda}$, that is, $I(\boldsymbol{\lambda}) = 0$ and $\partial I/\partial z_j = 0$ at $\mathbf{z} = \boldsymbol{\lambda}$. The Hessian matrix of the joint rate function at the minimum is linked to the diffusion matrix $\mathbf{D}$ as

$$\left.\frac{\partial^2 I}{\partial z_j \partial z_k}\right|_{\mathbf{z}=\boldsymbol{\lambda}} = Q_{j,k} = (\mathbf{D}^{-1})_{j,k}, \tag{45}$$

where here $\mathbf{D}$ denotes the $K \times K$ part of the diffusion matrix corresponding to the $K$ retained FTLEs. A second-order Taylor expansion of the joint rate function in the vicinity of $\boldsymbol{\lambda}$,

$$I(\mathbf{z}) \approx \frac{1}{2}(\mathbf{z} - \boldsymbol{\lambda})^{\mathrm{T}}\mathbf{Q}(\mathbf{z} - \boldsymbol{\lambda}), \tag{46}$$

corresponds to a multivariate Gaussian probability density with mean $\boldsymbol{\lambda}$ and covariance matrix $(\tau\mathbf{Q})^{-1}$, recovering the central limit theorem (CLT).

## 5.5 Estimating the joint rate function

There are again two ways of estimating the joint rate function from the time series of FTLEs: via the probability density function (cf., eq.(43)) or via the Legendre transform (cf., eq.(44)).

### 5.5.1 Probability density function approach

By inverting eq.(43) we get

$$I(\mathbf{z}) = -\lim_{\tau \to \infty} \frac{1}{\tau} \log p\left(\mathbf{\Lambda}^{(\tau)} = \mathbf{z}\right). \tag{47}$$

The probability density of $\mathbf{\Lambda}^{(\tau)}$ is modelled as

$$p\left(\mathbf{\Lambda}^{(\tau)} = \mathbf{z}\right) = \frac{1}{Z^{(\tau)}} \exp\left[-U^{(\tau)}(\mathbf{z})\right] \tag{48}$$

with normalisation constant

$$Z^{(\tau)} = \int_{\mathbb{R}^K} \exp\left[-U^{(\tau)}(\mathbf{z})\right] d^K \mathbf{z} \tag{49}$$

The potential function $U^{(\tau)}(\mathbf{z})$ is expanded into suitable multinomial basis functions as

$$U^{(\tau)}(\mathbf{z}) = \sum_{i=1}^{J} \beta_i^{(\tau)} \phi_i(\mathbf{z}) \tag{50}$$

subject to appropriate conditions to ensure a normalisable probability density. The expansion coefficients $\{\beta_i^{(\tau)}\}_{i=1}^{J}$ are determined from the time series of the FTLEs $\{\mathbf{\Lambda}_\alpha^{(\tau)}\}_{\alpha=0}^{L-1}$ via maximum likelihood which is a convex optimisation problem with unique solution.

The estimate of the joint rate function is

$$I(\mathbf{z}) = \lim_{\tau \to \infty} \frac{1}{\tau} \left[U^{(\tau)}(\mathbf{z}) - U^{(\tau)}(\mathbf{z}^*)\right] \tag{51}$$

where $\mathbf{z}^*$ denotes the position of the minimum of the potential function $U^{(\tau)}(\mathbf{z})$ which for finite $\tau$ is not necessarily equal to $\boldsymbol{\lambda}$.

### 5.5.2 Legendre transform approach

Alternatively, the joint rate function $I(\mathbf{z})$ can be determined via the multivariate Legendre transform of eq.(44) with the moment generating function estimated as the sample mean over the time series:

$$\left\langle e^{\boldsymbol{\theta}^{\mathrm{T}} \mathbf{\Lambda}^{(\tau)}} \right\rangle = \frac{1}{L} \sum_{\alpha=0}^{L-1} e^{\boldsymbol{\theta}^{\mathrm{T}} \mathbf{\Lambda}_\alpha^{(\tau)}} \tag{52}$$

Again, this is a convex optimisation problem and rate functions obtained from the Legendre transform method are guaranteed to be strictly convex with a unique zero and minimum at $\mathbf{z}^* = \boldsymbol{\lambda}$.

**Table 4.** Methods for estimating the rate function

| | Probability density | Legendre transform |
|---|---|---|
| Univariate | $I_j(z) = \lim\limits_{\tau \to \infty} \frac{1}{\tau}\left[ U_j^{(\tau)}(z) - U_j^{(\tau)}(z^*) \right]$ | $I_j(z) = \lim\limits_{\tau \to \infty} \frac{1}{\tau} \sup\limits_{\theta \in \mathbb{R}} \left[ \theta z - \log\left( \frac{1}{L} \sum_{\alpha=0}^{L-1} e^{\theta \Lambda_{j,\alpha}^{(\tau)}} \right) \right]$ |
| Multivariate | $I(\mathbf{z}) = \lim\limits_{\tau \to \infty} \frac{1}{\tau}\left[ U^{(\tau)}(\mathbf{z}) - U^{(\tau)}(\mathbf{z}^*) \right]$ | $I(\mathbf{z}) = \lim\limits_{\tau \to \infty} \frac{1}{\tau} \sup\limits_{\boldsymbol{\theta} \in \mathbb{R}^K} \left[ \boldsymbol{\theta}^{\mathrm{T}} \mathbf{z} - \log\left( \frac{1}{L} \sum_{\alpha=0}^{L-1} e^{\boldsymbol{\theta}^{\mathrm{T}} \boldsymbol{\Lambda}_\alpha^{(\tau)}} \right) \right]$ |

**Table 5.** Methods for estimating the diffusion matrix

| | Probability density | Legendre transform | Time series |
|---|---|---|---|
| Univariate | $D_{j,j}^{-1} = I_j''(z^*)$ | $D_{j,j}^{-1} = I_j''(\lambda_j)$ | $D_{j,j} = \lim\limits_{\tau \to \infty} \frac{\tau}{L} \sum_{\alpha=0}^{L-1} \left( \Lambda_{j,\alpha}^{(\tau)} - \lambda_j \right)^2$ |
| Multivariate | $(\mathbf{D}^{-1})_{j,k} = \left. \frac{\partial^2 I}{\partial z_j \partial z_k} \right|_{\mathbf{z}=\mathbf{z}^*}$ | $(\mathbf{D}^{-1})_{j,k} = \left. \frac{\partial^2 I}{\partial z_j \partial z_k} \right|_{\mathbf{z}=\boldsymbol{\lambda}}$ | $D_{j,k} = \lim\limits_{\tau \to \infty} \frac{\tau}{L} \sum_{\alpha=0}^{L-1} \left( \Lambda_{j,\alpha}^{(\tau)} - \lambda_j \right)\left( \Lambda_{k,\alpha}^{(\tau)} - \lambda_k \right)$ |

## 5.6 Estimating the diffusion matrix

The diffusion matrix $\mathbf{D}$ (or the part of it corresponding to the $K$ considered FTLEs) can be obtained from both joint rate function estimates as the inverse of the Hessian matrix at the minimum (cf., eq.(45)). It can also be estimated directly from the time series of the FTLEs as given in eq.(17). The estimates from the Legendre transform-based joint rate function and from the time series are always the same, apart from errors in the finite-difference approximation of the second derivatives. The diffusion matrix estimates from the probability density-based joint rate function and from the time series are the same if the model in eq.(50) is a multivariate Gaussian probability density; otherwise they are different.

The different methods for estimating the rate function and the diffusion matrix in the univariate and the multivariate case are summarised in Tables 4 and 5.

In high-dimensional systems it is usually too ambitious a task to determine $I(\mathbf{z})$ beyond the Gaussian approximation for the full system. We here restrict ourselves to the bivariate case $K = 2$.

## 6 Results

## 6.1 Lyapunov exponents

Time series of FTLEs of the QG model of length $L = 25000$ with a basic integration time $\Delta\tau = 1$ day are generated as described in Section 3. The (global) LEs are calculated as

$$\lambda_j = \frac{1}{L} \sum_{\alpha=0}^{L-1} \Lambda_{j,\alpha}^{(\Delta\tau)}. \tag{53}$$

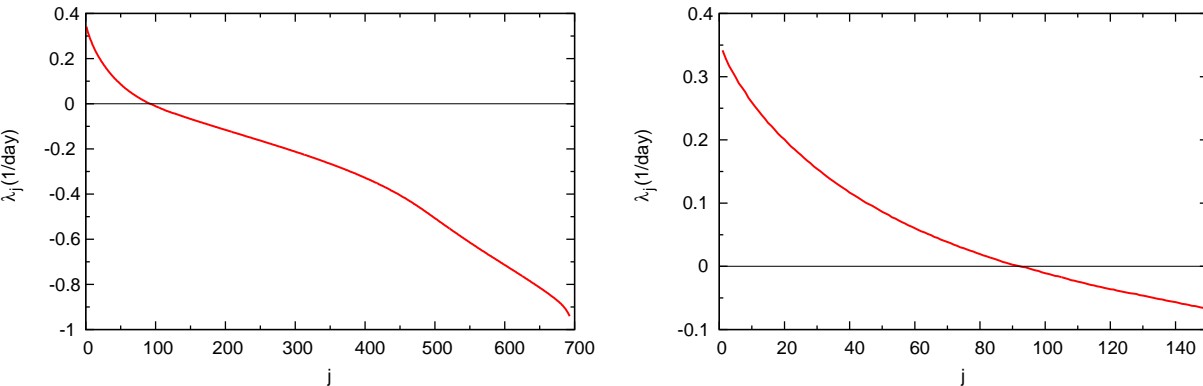

**Figure 1.** (a) Lyapunov spectrum of the QG model. (b) Close-up of (a).

Figure 1 displays the Lyapunov spectrum of the QG model. There are 91 positive LEs. The largest LE is estimated as $\lambda_1 = 0.344$/day, corresponding to an $e$-folding time of perturbation growth of 2.9 days which appears to be realistic for the real atmosphere. The spectrum starts off quite steep and then flattens at the near-zero exponents. For example, there are 69 LEs between 0.05/day and -0.05/day. The spectrum becomes steeper again at the trailing very stable exponents. Overall, there
is a continuous spectrum of timescales with no clear timescale separation. This is in accordance with previous results for QG models (Vannitsem and Nicolis, 1997; Schubert and Lucarini, 2015) and is probably because QG equations are scale-filtered equations.

Figure 2 shows the standard deviation $\sigma_j^{(\tau)}$ of the fluctuations of the FTLEs around their mean values (eq.(40)). The standard deviation monotonically decreases with increasing integration time $\tau$ for all exponents. The fluctuations are largest for the
leading LEs and then quickly decrease. They increase again towards the end of the Lyapunov spectrum with a particularly sharp increase for the most stable exponents at the very end of the spectrum. This is in line with similar findings in simple spatially extended systems (Kuptsov and Politi, 2011; Pazó et al., 2013) as well as in a QG atmosphere-ocean model (Vannitsem and Lucarini, 2016).

The scaled standard deviation $\sigma_j^{(\tau)}\tau^{1/2}$ shows clear convergence for all of the exponents at $\tau = 10 - 15$ days, that is, the
scaled variance converges to the diagonal elements $D_{j,j}$ of the diffusion matrix $\mathbf{D}$. Convergence is reached at about $\tau = 10$ days for almost all of the exponents; it is particularly fast for the near-neutral and the weakly dissipative exponents where it is reached already at $\tau = 5 - 10$ days.

There is a kink-like feature at $j \approx 125$, separating regions with different slopes of the standard deviation. It is possible that this is linked to a distinction of the covariant Lyapunov vectors into interacting 'physical modes' and hyperbolically separated
~~dissipative~~ 'isolated modes'~~, and thus to the dimension of the inertial manifold or the effective number of degrees of freedom of the system~~ (Yang et al., 2009). But this certainly needs further investigation.

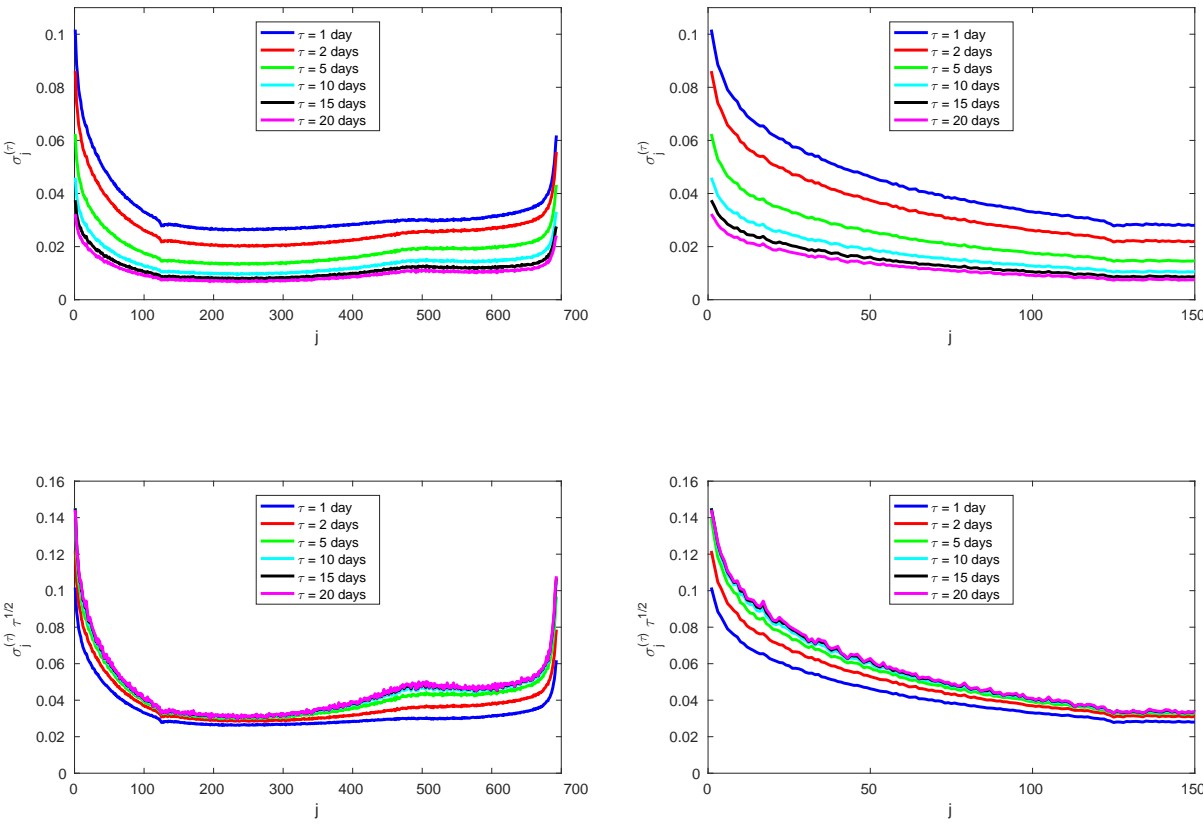

**Figure 2.** (a) Standard deviation $\sigma_j^{(\tau)}$ of the FTLEs. (b) Close-up of (a). (c) Scaled standard deviation $\sigma_j^{(\tau)}\tau^{1/2}$ of the FTLEs. (d) Close-up of (c).

## 6.2 Multivariate fluctuation analysis

Figure 3 shows the explained variance and the cumulative explained variance of the principal components of the scaled Lyapunov fluctuations. There are three leading modes, then the eigenvalue spectrum sharply flattens off. The fraction of variance explained by the leading modes increases with increasing integration time $\tau$. Going from $\tau = 1$ day to $\tau = 20$ days, the variance explained by the first principal component increases from just below 5% to more than 12%, and the variance explained by the second principal component increases from about 2% to more than 4%. However, due to the flatness of the bulk of the eigenvalue spectrum, even in the diffusion limit a substantial number of modes is necessary to explain large parts of the fluctuation variance. The eigenvalue spectrum is still not fully converged at $\tau = 20$ days. It is not completely clear what the reason for this is. There may be some indication that the off-diagonal elements of the diffusion matrix converge slightly more

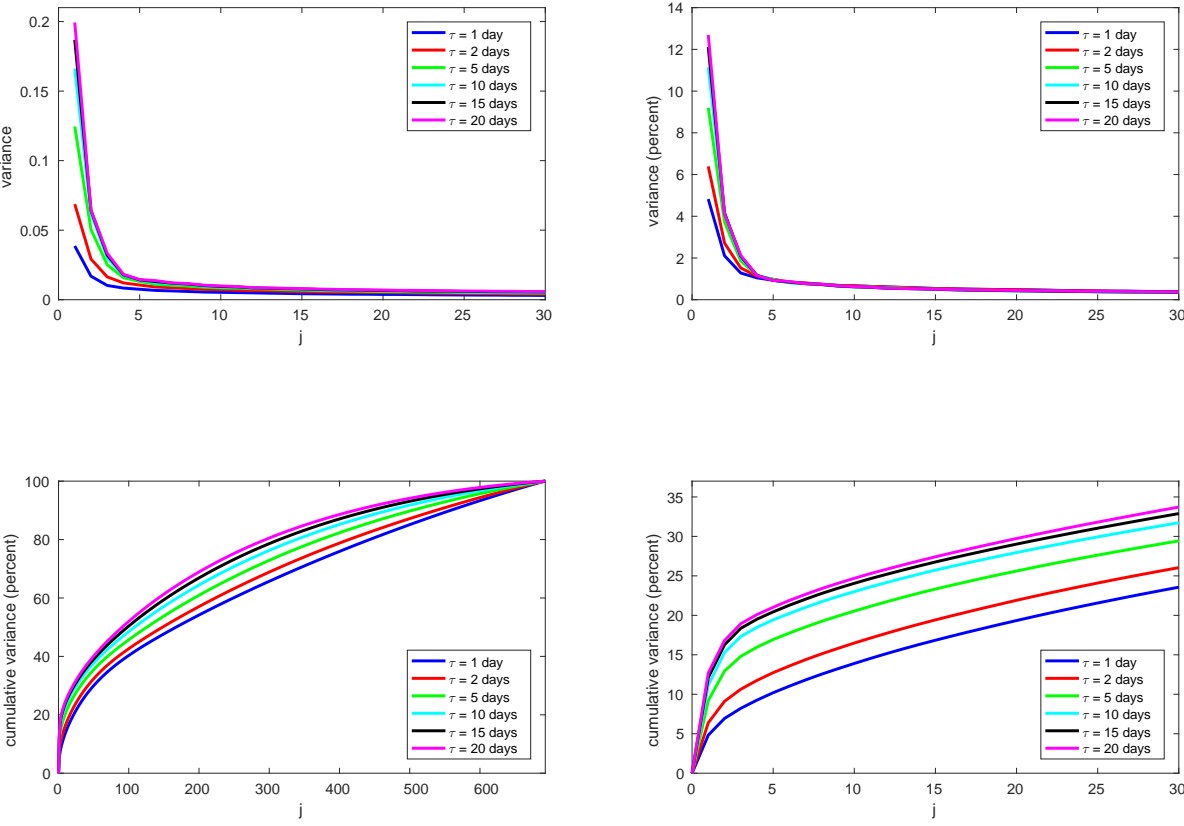

**Figure 3.** (a) Variance of the principal components of the finite-time Lyapunov fluctuations. (b) Fraction of variance. (c) Cumulative fraction of variance. (d) Close-up of (c).

slowly than the diagonal elements. But there is probably also a finite sample size effect. With increasing $\tau$, the time series of the FTLEs contain less and less uncorrelated information and fail to fully sample the high-dimensional covariance matrix which leads to an overestimation of the variance of the leading principal components.

In Figure 4 the three leading EOFs are displayed. The modes are largely independent of the integration time $\tau$ and have converged at about $\tau = 10$ days. The first EOF shows a pattern where all of the leading FTLEs fluctuate in phase. This incorporates all of the positive exponents and extends to the weakly dissipative ones. Then there is some negative correlation with the dissipative exponents in the second half of the Lyapunov spectrum. In the second EOF again the leading FTLEs fluctuate in phase; this here encompasses about the first 40 exponents. Then there is some negative correlation with the weakly dissipative exponents and substantial positive correlation with the strongly dissipative exponents at the end of the Lyapunov spectrum.

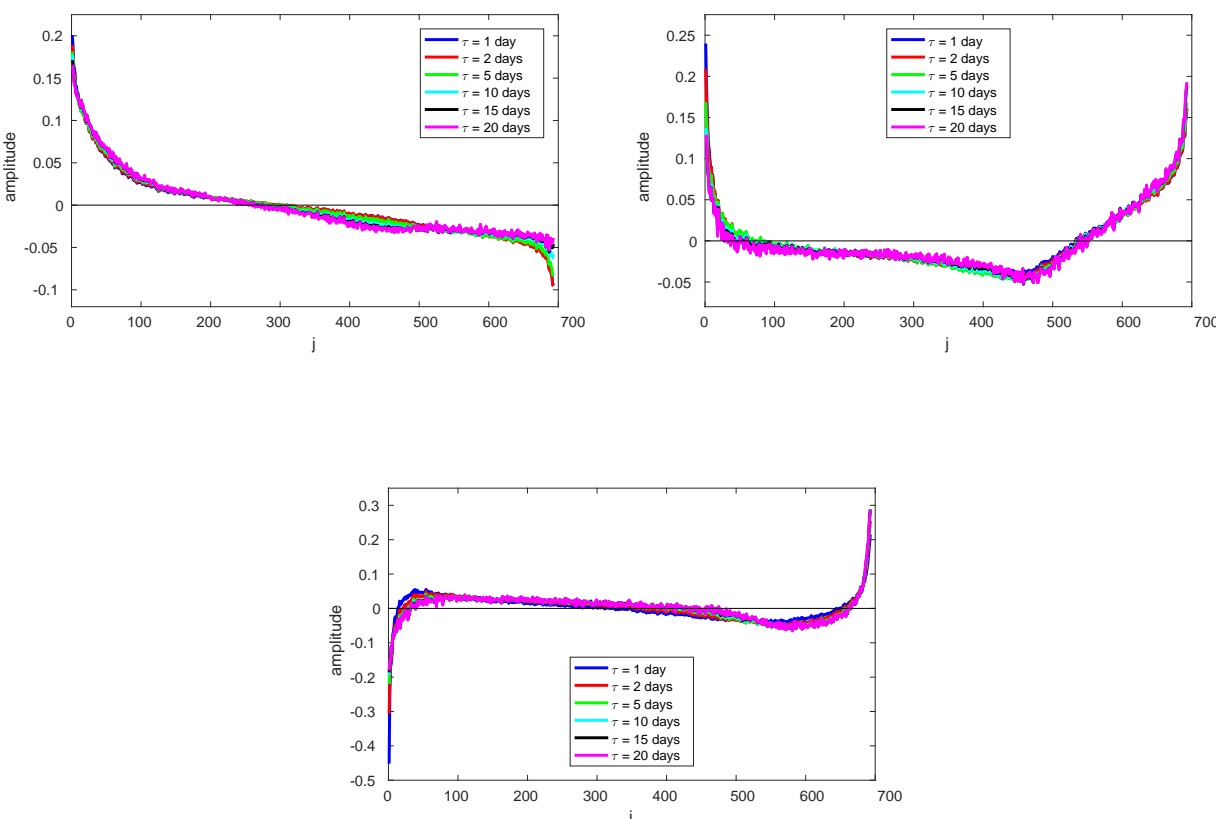

**Figure 4.** (a) First, (b) second and (c) third empirical orthogonal function (EOF) of the finite-time Lyapunov fluctuations.

The third EOF has the very stable exponents at the end of the spectrum fluctuating in phase and the most unstable exponents fluctuating in phase with each other, out of phase with the dissipative ones.

Complementary to the EOF analysis, Figure 5 shows the correlation of selected FTLEs with each of the other FTLEs for $\tau = 1$ day and $\tau = 15$ days. The pattern of the correlations is the same for both integration times but the amplitudes are very low for $\tau = 1$ day and build up at larger integration times. This is in line with the results from the EOF analysis. The FTLEs have predominantly positive correlations with neighbouring exponents; these are strongest for the most unstable and the most stable exponents and weaker in between. There are also some relatively weak long-range correlations across the Lyapunov spectrum.

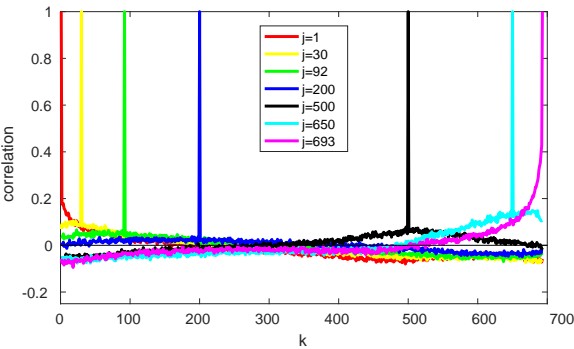 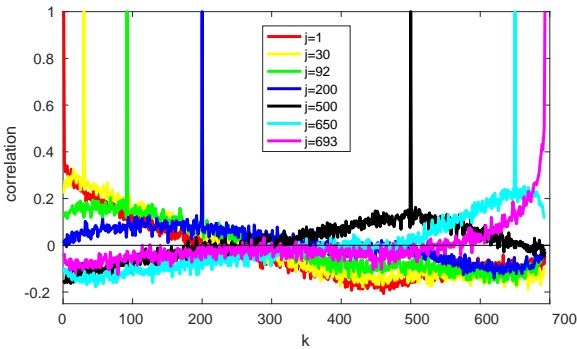

**Figure 5.** Correlation of the FTLEs $\Lambda_j^{(\tau)}$ and $\Lambda_k^{(\tau)}$ for (a) $\tau = 1$ day and (b) $\tau = 15$ days.

### 6.3 Large-deviation analysis

#### 6.3.1 One-dimensional approach

We now investigate whether the fluctuations of the FTLEs obey a large deviation principle. As representative examples we look at the first and the fifth exponent as two strongly unstable modes, at the zero exponent, at a weakly dissipative exponent and at the smallest, most stable exponent. The large-deviation rate function is estimated as described in Section 5 from the probability density function and via the Legendre transform for various values of $\tau$. The corresponding element $D_{j,j}$ of the diffusion matrix is calculated from the curvature of the two estimates of the rate function and directly from the time series of the FTLEs.

To model the probability density of the FTLEs two different choices for the potential function in eq.(39) are considered here: $M = 2$, that is, a Gaussian probability density and $M = 4$, a fourth-order polynomial. In view of the high degree of correlation in the time series of the FTLEs, particularly for large $\tau$, model selection is here performed as follows. For $\tau = n\Delta\tau$, the time series of the FTLEs, $\{\Lambda_{j,\alpha}^{(\tau)}\}_{\alpha=0}^{L-1}$, are ~~split~~ subsampled into $n$ disjoint ~~subsets~~ time series with non-overlapping integration time intervals, $\{\Lambda_{j,m}^{(\tau)}, \Lambda_{j,m+n}^{(\tau)}, \Lambda_{j,m+2n}^{(\tau)}, \ldots\}$, for $m = 0, \ldots, n-1$. The length of the subsampled time series is the largest integer $L'$ such that $m + (L'-1)n \leq L-1$. The two probability density models are fitted separately on the ~~subsets~~ $n$ subsampled time series and model selection is based on the average Bayesian information criterion ~~over the subsets~~. Then the selected model is fitted on the whole time series.

Figure 6 displays the order of the model for the probability density of the selected FTLEs as a function of the integration time $\tau$. The leading unstable exponents exhibit strong non-Gaussianity. For the first exponent, it is detectable up to $\tau = 35$ days; for the fifth exponent, it is less pronounced and visible only up to $\tau = 12$ days. The zero exponent shows only very mild non-Gaussianity which is visible for $\tau = 1$ day and $\tau = 2$ days. The weakly dissipative exponent has Gaussian behaviour at

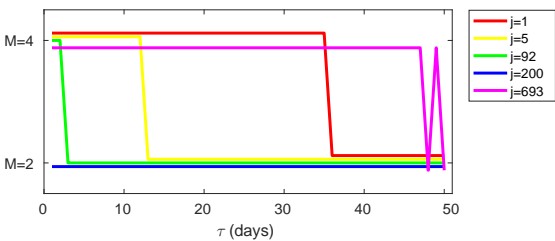

**Figure 6.** Order of the model for the probability density function of the FTLE $\Lambda_j^{(\tau)}$.

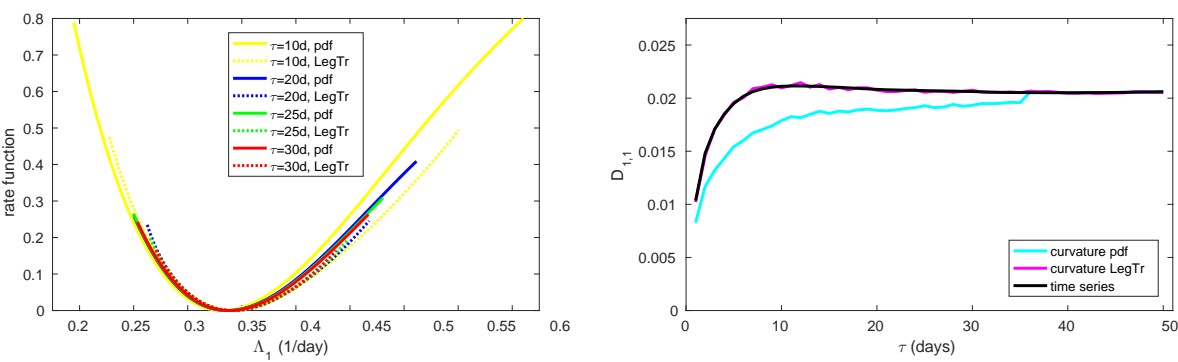

**Figure 7.** (a) Large-deviation rate function of the first FTLE. (b) Element $D_{1,1}$ of the diffusion matrix.

all values of $\tau$. The smallest, strongly dissipative exponent again displays marked deviations from Gaussianity; these are even more pronounced than those for the first exponent and detectable up to an integration time as large as $\tau = 49$ days. For the first and the last exponent, at small integration times $\tau$ it may be possible to even switch to the higher-order model $M = 6$ but this is not our concern here.

5    Figure 7 shows the results of the large-deviation analysis for the first FTLE. Convergence to a large-deviation principle is observed. At $\tau = 10$ days and even visible at $\tau = 20$ days the maximum of the probability density is still shifted away from the mean; nevertheless, some convergence among the probability density-based estimates of the rate function is reached at about

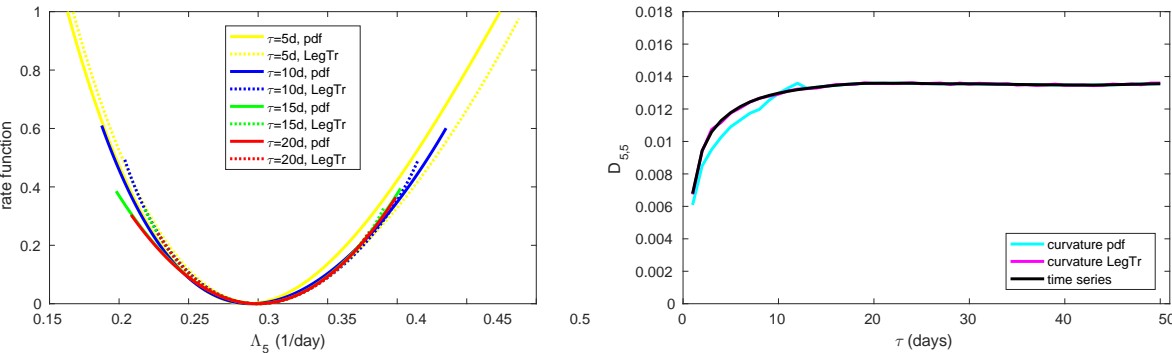

**Figure 8.** (a) Large-deviation rate function of the fifth FTLE. (b) Element $D_{5,5}$ of the diffusion matrix.

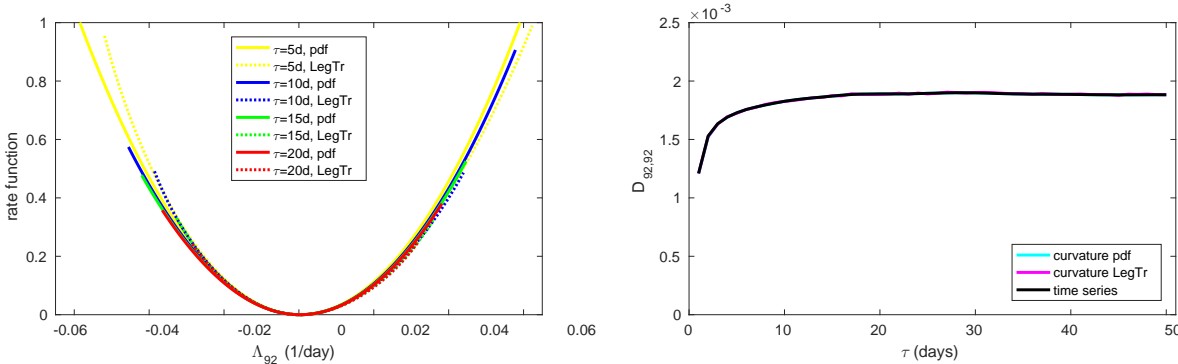

**Figure 9.** (a) Large-deviation rate function of the 92th FTLE. (b) Element $D_{92,92}$ of the diffusion matrix.

$\tau = 20$ days. The Legendre transform-based estimates give a consistent picture already from $\tau = 10$ days. Good convergence is also observed for the corresponding element of the diffusion matrix.

For the fifth FTLE, a similar picture can be seen (Figure 8) but convergence is markedly faster than for the first FTLE. The probability density-based estimates are very consistent from $\tau = 10 - 15$ days; note that the model for the probability density jumps from fourth-order to Gaussian for the higher values of $\tau$. The Legendre transform gives close agreement for the rate function already from $\tau = 5$ days.

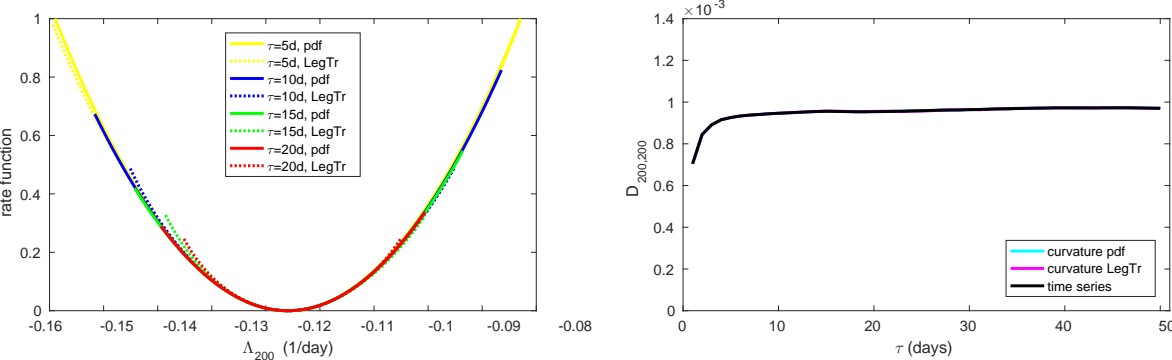

**Figure 10.** (a) Large-deviation rate function of the 200th FTLE. (b) Element $D_{200,200}$ of the diffusion matrix.

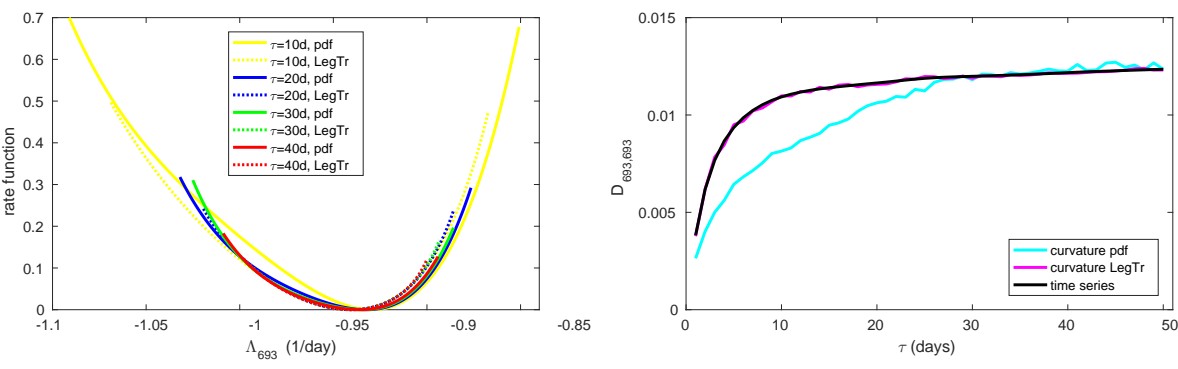

**Figure 11.** (a) Large-deviation rate function of the 693th FTLE. (b) Element $D_{693,693}$ of the diffusion matrix.

For the zero exponent (Figure 9), convergence is again markedly faster than for both positive exponents. A large-deviation principle can be established already from about $\tau = 10$ days and the two different estimates of the rate function are close together. The estimates of the diffusion coefficient all coincide.

For the fully Gaussian 200th FTLE (Figure 10), convergence is even faster. A large-deviation principle is valid from $\tau = 5$ days and all of the estimates of the rate function are in almost perfect agreement. The estimates of the diffusion coefficient show corresponding behaviour.

For the smallest, most dissipative exponent (Figure 11), the convergence to a large-deviation principle is very slow, even slower than for the first, most unstable exponent. A large-deviation principle is valid from about $\tau = 30$ days and the Legendre

**Table 6.** Correlation length $l_j^{(\Delta\tau)}$ of the time series of the FTLE $\Lambda_j^{(\Delta\tau)}$ with $\Delta\tau = 1$ day

| $j$ | 1 | 5 | 92 | 200 | 693 |
|---|---|---|---|---|---|
| $l_j^{(\Delta\tau)}$ | 2.07 | 2.10 | 1.61 | 1.38 | 3.25 |

transform method gives reliable estimates of the rate function from $\tau = 10 - 20$ days. Also the convergence of the diffusion coefficient is markedly slow. The estimate from the non-Gaussian probability density is initially too low and converges at about $\tau = 25$ days.

The different speeds of convergence to a large-deviation principle for the different FTLEs can be understood from the degrees of serial correlation and non-Gaussianity of the FTLEs. The correlation length of the FTLE $\Lambda_j^{(\tau)}$ is defined as

$$l_j^{(\tau)} = 1 + 2\sum_{i=1}^{\infty} \rho_{j,i}^{(\tau)} \tag{54}$$

where $\rho_{j,i}^{(\tau)}$ is the autocorrelation of the FTLE $\Lambda_j^{(\tau)}$ at lag $i$. Note that a lag of 1 here refers to two consecutive but non-overlapping integration time intervals of length $\tau$, that is, $\rho_{j,i}^{(\tau)}$ is the autocorrelation at lag $i$ of the subsampled time series of $\Lambda_j^{(\tau)}$ as introduced above for the model selection of the probability density model, $\{\Lambda_{j,m}^{(\tau)}, \Lambda_{j,m+n}^{(\tau)}, \Lambda_{j,m+2n}^{(\tau)}, \ldots\}$, for $m = 0, 1, \ldots, n-1$. There are $n$ of these which can be used to generate $n$ estimates of $l_j^{(\tau)}$ and then take the average. The definition of correlation length of equation (54) occurs naturally in the formulation of the CLT for dependent random variables (e.g., Billingsley, 1995) under the assumption of a Markov process which is sufficiently mixing. Consider now two integration times $\tau_1 = n_1\Delta\tau$ and $\tau_2 = n_2\Delta\tau$ with $n_1 \leq n_2$ and $n' = n_2/n_1$ being an integer for simplicity; one could consider a continuous integration time $\tau$ in the limit $\Delta\tau \to 0$. The variances of $\Lambda_j^{(\tau_1)}$ and $\Lambda_j^{(\tau_2)}$ are linked as $\left[\sigma_j^{(\tau_2)}\right]^2 = \left[\sigma_j^{(\tau_1)}\right]^2 l_j^{(\tau_1)}/n'$ and the two estimates of the diffusion coefficient as calculated from the time series or the Legendre transform are linked as $D_{j,j}^{(\tau_2)} = D_{j,j}^{(\tau_1)} l_j^{(\tau_1)}$. This holds in the limit $n' \to \infty$, otherwise $l_j^{(\tau_1)}$ needs to be replaced with a counterpart which takes only a finite number of lags into account and also contains some correction terms. Convergence to a large-deviation principle is limited by serial correlation of the FTLEs. Convergence to the diffusion limit, that is, to the Gaussian approximation of the large-deviation regime can certainly not be expected before the serial correlations have decayed, that is, when $l_j^{(\tau)} \approx 1$. If the distribution of the FTLEs is Gaussian or close to Gaussian the large-deviation limit is equivalent to the diffusion limit and convergence occurs immediately after correlation decay; otherwise it is further delayed, generally the longer the larger the departure from Gaussianity is.

Table 6 gives the correlation length of the selected FTLEs $\Lambda_j^{(\Delta\tau)}$ at the basic integration time $\Delta\tau = 1$ day. Note that $l_j^{(\Delta\tau)}$ does not allow to directly calculate the value of $\tau$ at which convergence to a large-deviation principle occurs but it gives an impression of the timescales of temporal correlation and how they differ for the different FTLEs. Overall, temporal correlation is not very pronounced for all of the FTLEs but the correlation length varies by a factor of 2.35 from the shortest to the longest. The rapid convergence to a large-deviation principle for the zero and weakly dissipative exponents is in line with their short

correlation length and almost Gaussian distribution. For the first and the last exponent, due to the strong non-Gaussianity convergence is delayed beyond what is expected from the somewhat larger correlation length.

### 6.3.2 Two-dimensional approach

As an example of a multivariate large-deviation analysis, Figure 12 shows the joint large-deviation rate function of the first two FTLEs, $\Lambda_1$ and $\Lambda_2$. The estimates of the diffusion coefficients $D_{1,1}$, $D_{2,2}$ and $D_{1,2}$ are also shown. The potential function for the joint probability density is chosen as

$$U^{(\tau)}(z_1, z_2) = \sum_{i=1}^{M} \sum_{j=0}^{i} \beta_{i,j}^{(\tau)} \left( \frac{z_1 - \lambda_1}{\sigma_1^{(\tau)}} \right)^{i-j} \left( \frac{z_2 - \lambda_2}{\sigma_2^{(\tau)}} \right)^{j} \tag{55}$$

where the order of the model is fixed a priori at $M = 4$. The joint rate function displays markedly non-Gausian behaviour and some dependence between $\Lambda_1^{(\tau)}$ and $\Lambda_2^{(\tau)}$. Convergence to a large-deviation principle is mainly reached at $\tau = 15$ days as can be seen from the probability density-based estimates of the joint rate function. The estimates from the Legendre transform are in agreement and indicate the joint rate function already at $\tau = 10$ days. The elements of the diffusion matrix are overall well estimated with detailed convergence being somewhat slow in accordance with the univariate analysis for the first FTLE. The estimate of the off-diagonal element $D_{1,2}$ is particularly good.

## 7 Conclusions

The statistical properties of the fluctuations of FTLEs were investigated in a three-level quasi-geostrophic atmospheric model with realistic mean state and variability. The Lyapunov spectrum of the model has almost 100 positive LEs and displays no clear timescale separation.

A principal component analysis of the fluctuations of the FTLEs around their mean values was performed. The scaled covariance matrix of the fluctuations is converged to the limiting diffusion matrix at about $\tau = 15$ days. There are substantial correlations among the different FTLEs. The first three empirical orthogonal functions are patterns where the leading positive FTLEs fluctuate together in phase. These modes are largely independent of the integration time $\tau$.

A large-deviation principle can be established for all of the FTLEs. The convergence to the large-deviation limit behaviour is slightly slow for the most unstable and the most stable FTLEs and very fast in between. Convergence to the diffusion limit, that is, to the Gaussian approximation of the large-deviation regime is generally faster. Also a joint large-deviation rate function for the first and the second FTLE was successfully estimated beyond the Gaussian approximation. Good correspondance was found between the curvature of the rate functions at the minimum and the corresponding elements of the diffusion matrix.

Two different methods for estimating the large-deviation rate functions from the data were discussed: an approach via the probability density function and an approach using the Legendre transform. The Legendre transform method appears to be generally superior for finding the rate function as (i) convergence occurs at a smaller value of the integration time $\tau$ where more independent data are available and (ii) it yields diffusion coefficients fully consistent with their direct estimation from

the data. Nevertheless, both methods should be considered side by side as the probability density approach allows to monitor if/when the probability density function has actually reached the large-deviation regime.

*Competing interests.* The author declares that they have no conflict of interest.

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

# Responses to Reviewers' Comments

The author would like to thank both reviewers very much for their useful comments which helped further improve the manuscript.

## Reviewer 1

– A. Large deviation theory

Q:

The presentation of the large deviation theory is a mess, specially considering the presentation of the block averaging method. Is not enough to say ...

A:

The presentation of the large-deviation theory has been simplified as suggested by the reviewer (see revised manuscript).

– B. Inertial manifold

Q:

At the end of Sec. 6.2 the new paragraph on the inertial manifold is not correct. It points to a dimensionality of the inertial manifold ($\approx 125$) much smaller than the dimensionality of the attarctor obtained by the Kaplan-Yorke formula. This is not possible: the dimension of the attractor is necessarily smaller than the dimension of the inertial manifold.

A:

Yes, the author is aware of this. It is not claimed that the two dimensions are the same; it is just stated that there may be a relationship between the standard deviations of the FTLEs and the existence of entangled and isolated Lyapunov modes which needs to be investigated further. This is made clearer in the revised manuscript.

– C. Correlation length

1. Q:

The formula for the correlation length in Eq.(57) is puzzling. Where it comes from? What hypotheses have been assumed?

A:

The expression for the correlation length occurs in the formulation of the central limit theorem for dependent random variables based on the assumption of a sufficiently mixing Markov process. This information is given in the revised manuscript.

2. Q:

The text suggests that the crossover $\tau$ for convergence to the large-deviation regime is related to the correlation time of the FTLE. I disagree with this interpretation for two reasons. First, the correlation lengths observed are not much different in comparison with the crossover $\tau$'s. Second, in spatially extended systems it is typical that

correlation times are insensitive to the system size, while crossover $\tau$ scales with the system size as $L^z$, $z > 0$. This indicates a different operate in each case (Pazó et al., 2013).

A:

The author thinks the crossover $\tau$ for convergence to the large-deviation regime is determined by both the correlation length and the degree of non-Gaussianity of the distribution of the FTLEs. This is made more precise and linked explicitly to the definition of the correlation length in the revised manuscript. The severely delayed convergence for the first and the last exponent is due to the strong non-Gaussianity of these FTLEs. The dependence of the crossover $\tau$ on the system size is not discussed in the manuscript.

– D. Legendre transform method

Q:

I encourage the author to emphasize in the conclusions that the Legendre transform approach appears to be superior to the probability density approach in order to find the rate function (assumed the conditions of the Gärtner-Ellis theorem hold). Moreover the Legendre transform method yields diffusion coefficients fully consistent their direct measurement from the data.

A:

Some remarks along these lines have been added to the conclusions.

– E. Figures

1. Q:

In Figs. 2(c) and 2(d) the factor $\tau^{1/2}$ should appear in the $y$-axis label.

A:

Done.

2. Q:

I have a suggestion, which can be followed or not: In Figs. 3(a) and 3(b), wouldn't it be clearer to set log scale for the $y$-axis?

A:

I tried this but it appears not to be really helpful.

3. Q:

Figure 4 is more clear writing $y_j^{(\tau)}$ in the $y$-axis label.

A:

$y_j^{(\tau)}$ is the amplitude of the principal component but Figure 4 shows the amplitude of the eigenvectors $\mathbf{e}_j^{(\tau)}$.

4. Q:

Contrary to what is claimed in the author's response, the labels '(a)', '(b)', etc. are not included in the revised ms.

A:

I think it will be more convenient to put the labels in during typesetting of the manuscript as some of these will need to be placed outside of the figures (e.g., for Figure 12).

– F. Typos

1. Q:

   Page 17, line 8, $n \rightarrow L/n$.

   A:

   There are actually $n$ subsets (or subsampled time series) and the length of each is $L/n$ (or more precise the largest integer $L'$ such that $m + (L' - 1)n \leq L - 1$. This is clarified in the revised manuscript.

2. Q:

   Page 21, line 4, to $\rightarrow$ too.

   A:

   Corrected.

## Reviewer 2

– Q:

Line 10, page 2, the author indicates that the large deviation theory will be used to characterize the variability of the FTLEs. I suggest to introduce there a paragraph explaining what is the interest of the large deviation theory to characterize the FTLEs. Also some comments on the usefulness of the approach in line with the results obtained in the manuscript would be worth doing in the concluding section. This should help the reader who is not familiar with the approach to apprehend the importance of the technique.

A:

Some comments along these lines have been included in the revised manuscript.

– Q:

Page 9, line 11, 2 references to Kwasniok (2019a, 2019b) are mentioned. When looking at the list of references, it appears that these works are in progress and are not effective references. This is not accepted by the journal. So these references have to be removed (if not accepted yet), and a possible solution is to provide a brief summary of the technicalities that were referred to in an Appendix. This can be short.

A:

The references have been removed.

– Q:

Line 11 of page 13, please remove the reference to Vannitsem and Lucarini (2016). This is not referred at the appropriate place.

A:

The reference has been removed at this place.

– Q:

Please check the labels in the panels of the different figures. Many are missing, if not all.

A:

I think it will be more convenient to put the labels in during typesetting of the manuscript as some of these will need to be placed outside of the figures (e.g., for Figure 12).

– Q:

Figure 7. Please clarify what is LegTr.

A:

LegTr refers to the Legendre transform method and this is clarified now.

– Q:

Line 4, page 21. "to" should be "too".

A:

Corrected.

– Q:

Page 23, line 15. Please remove the references to Kwasniok 2019a, 2019b.

A:

The references have been removed.

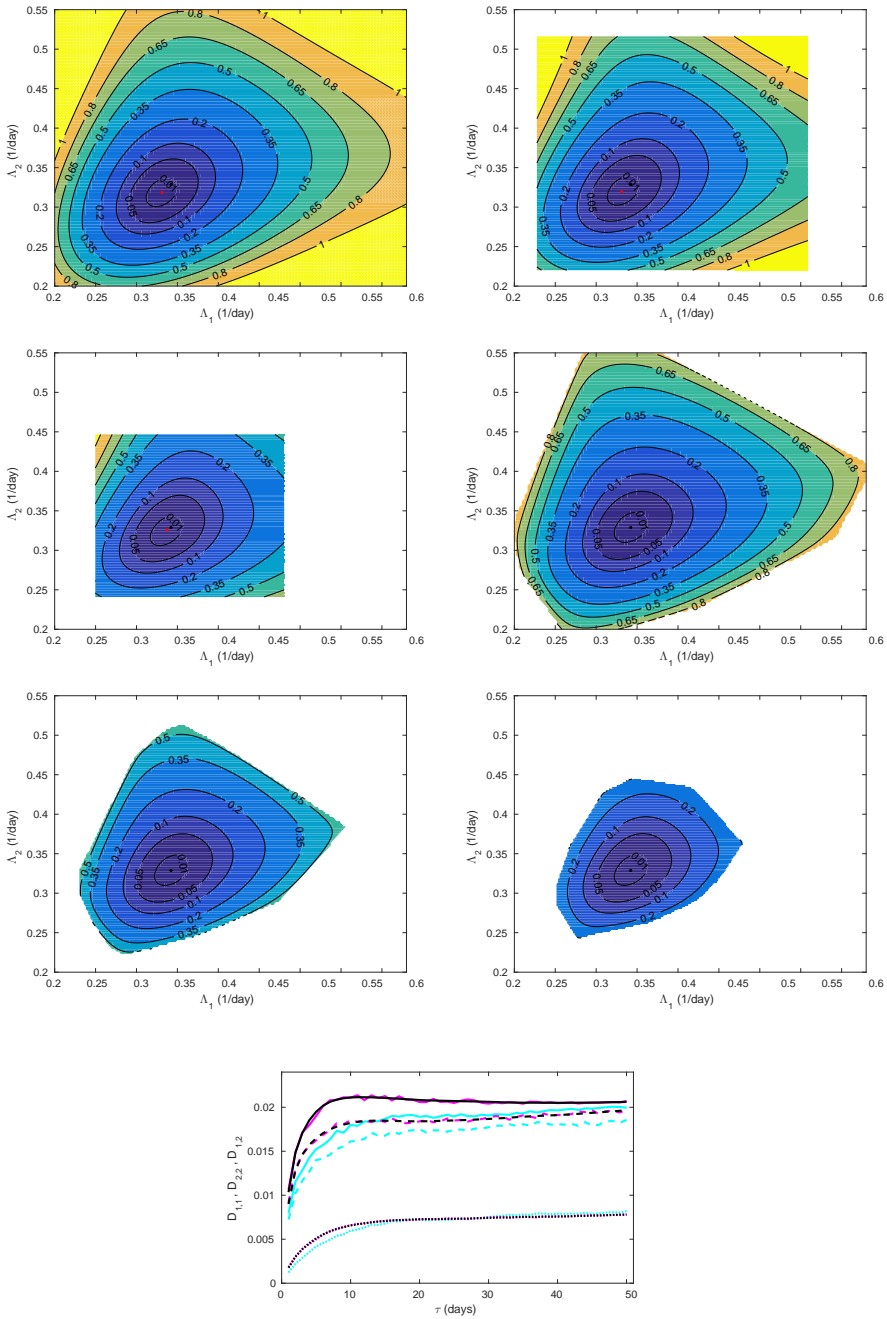

**Figure 12.** Joint large-deviation rate function of the first two FTLEs as estimated from the joint probability density for (a) $\tau = 10$ days, (b) $\tau = 15$ days and (c) $\tau = 25$ days; and with the Legendre transform for (d) $\tau = 10$ days, (e) $\tau = 15$ days and (f) $\tau = 25$ days. Black dots indicate the global LEs $(\lambda_1, \lambda_2)$; red dots in the panels (a), (b) and (c) indicate the maximum of the joint probability density. (g) Elements $D_{1,1}$ (solid), $D_{2,2}$ (dashed) and $D_{1,2}$ (dotted) of the diffusion matrix as estimated from the curvature of the probability density-based rate function (cyan), from the curvature of the Legendre transform-based rate function (magenta) and from the time series of the FTLEs (black).