# Peer review of "Fluctuations of finite-time Lyapunov exponents in an intermediate-complexity atmospheric model: a multivariate and large-deviation perspective"

_Nonlinear Processes in Geophysics, 2018_

## Referee Comment (RC1) · Anonymous Referee #1 · 25 May 2018

This manuscript investigates the fluctuations of the finite-time Lyapunov exponents in a three level quasi-geostrophic model. Fluctuations as well as the correlations of these fluctuations are analysed by means of empirical orthogonal functions and the large-deviations formalism. This work constitutes a relevant contribution to the general question of how "chaoticity" fluctuates, which can be potentially interesting for forecasting purposes. The scientific content is novel.

My main criticism to this work is its confusing presentation, which seriously hinders the readability and the appeal of this work. My judgement is that a substantial revision of

the manuscript is needed prior to publication.

I have several technical questions that I organize by sections:

* Section 2

In the model, if I correctly understand, the "h" term is only a parameter (not a function). If so, I think the model is invariant under a zonal shift. This neutral transformation would imply a second vanishing Lyapunov exponent in the spectrum. Please, clarify this point.

* Section 6.1

Concerning the last sentence in Sec. 6.1. I note that the equivalence between Lyapunov exponent fluctuations measured from Gram-Schmidt vectors and from covariant vectors, was detected already in Figs. 5 and 7 of Ref. [1].

In fact, the large fluctuations observed at the edges of the spectrum are not really surprising, at the light of the previous results on the diffusion coefficients in (Kuptsov and Polity, 2011) and [1].

* Section 6.2

It is absolutely necessary to include one formula defining the fraction of explained variance, in order to ensure the self-consistency of the text.

* Section 6.3

It is not said which is the total length of the time series used.

The value of $\tau_r$ is "hidden" in Sec. 3.

The three methods used to measure $D_{j,j}$ are not fully clear to me. I think the author should make a list with the three methods specifying which formulas are used in each one. And which parameters are used. Now the explanation is hidden in the caption of Fig. 5, and is hardly understandable.

After reading it several times I'm not sure if I correctly understand: Method one (red line) is computed directly from Eq. (14). The other two methods use the curvature of the rate function at its maximum. In one case (green line) the rate function is interpolated following Eq. (26) for several $\tau$ values. In any case, it is clear to me that this cannot be so good as method 1, because of the interpolations needed (details of this are unfortunately skipped). The last method (black line) uses the rate function in Eq. (29), which depends on previously estimated correlation lengths. All this information should be presented in a much more clear fashion. Now it is a mess.

———

Apart from the previous questions I have other minor recommendations/typos, (but I encourage the author to implement any other improvements in the presentation he may think of):

1. Two lines after Eq. (13), I would write scalar product instead of norm, because this is what matters for Gram-Schmidt orthogonalisation. I suppose the energy norm is trivially related to the scalar product used.

2. Vectors should be always typed in bold face, also for Greek letters.

3. Orthogonality of the eigenvectors, Eq. (18), is better written after Eq. (16).

4. The equation in the text preceding Eq. (18) is apparently lacking of $-\lambda$.

5. When introducing Eq. (27), it would be important to cite at least (Touchette, 2009) again and to mention this is the Gärtner–Ellis Theorem (if I'm not wrong).

5. The "log" symbol is missing in Eq. (35).

6. Figures 4-7 should be introduced in the text, one by one.

7. Last line of page 10. $\tau_c$ has not been defined.

8. Page 12, line 5. I don't appreciate smaller deviations of the rate function in this case

than for the zero exponent.

9. In figures 5-8, I would use different colours for the lines in panel (b), at least for the coloured ones since they are not related to the same colours in panel (a).

10. In the conclusions, it is mentioned that the most unstable exponents exhibit slower convergence to the large-deviation limit. Let me to point out that this is fully consistent with [1].

11. Labels (a), (b), etc. need to be included all the figures. This is critical in Fig. 9.

––––––––––––––––––––––––––––––

[1] Pazo et al, Phys. Rev. E 87, 062909 (2013).

––––––––––––––––––––––––––––

---

## Referee Comment (RC2) · Anonymous Referee #2 · 31 May 2018

In this manuscript the fluctuations of finite time lyapunov exponents are studied in an meteorologically relevant system. The author uses an interesting approach for studying collective excitations observed within the Lyapunov spectrum. I think the work is relevant for understanding the properties of fluctuations in a high dimensional dynamical system. Nevertheless, I think there should still be some improvement of presentation and explanation of the results.

On section 1: - Line 8: based from what I understand: Could the covariance structure tell us something about how "close" or how "interactive" the various unstable and stable

directions are? Could the covariance structure be related to the investigations of the inertial manifold using the angles between Lyapunov vectors. Maybe the work along the lines of Yang et al (2009) should be referenced here as a motivation.

On Section 2:

- I think a short concise table listing all paramerters of the model with their dimensional and a dimensional values would be beneficial to introduce the model setup.

On Section 3:

- It should be noted that the mean of the finite time Lyapunov exponents are in fact average growth rates of linear perturbations of the system. But the finite time LEs are not directly the growth rate of those perturbation. In fact one can define backward, forward and covariant LEs. There is a good review paper on this by Kuptsov and Parlitz (https://www.researchgate.net/publication/51961547_Theory_and_Computation_of_Covariant_Lyapunov_Vectors) which explains this distinction. I think using the FTLEs of the Gram Schmidt algorithm is alright, but it should be better clarified what type of FTLEs they actually are.

On Section 4:

I think this section should motivate better why one should use EOFs and what would be potentially alternatives to this approach.

On Section 6:

Section 6.1: - Since the model is zonally symmetric there should be two zero exponents. Can you verify this and could you include this in this discussion? - Figure 2: This result should be referenced with the findings about fluctations of the LE for covariant, backward and forward exponents in Vannitsem, Lucarini (2016). I think when you study collective excitations this is an interesting different viewpoint.

Section 6.2: - I think it would be helpful to present the matrix D as well as a surface plot and also use the first EOF and second EOF to see what parts of the D matrix are

actually reconstructed using the EOF method.

Section 7:

- That no clear time scale separation is found is probably because QG equations are scale filtered equations. Similar results where found before in QG models (Vannitsem 1997, Schubert 2015, Vannitsem 2016 ).

- Can the collective excitations be traced to any anomolous behavior in the non linear background state x? I think that would an interesting addition but of course not necessary in order to do this study.

References:

Vannitsem, S. and Lucarini, V. (2016) Statistical and dynamical properties of covariant lyapunov vectors in a coupled atmosphere-ocean model—multiscale effects, geometric degeneracy, and error dynamics. Journal of Physics A: Mathematical and Theoretical, 49 (22). 224001. ISSN 1751-8113

"Hyperbolicity and the Effective Dimension of Spatially Extended Dissipative Systems" Hong-liu Yang, Kazumasa A. Takeuchi, Francesco Ginelli, Hugues Chaté, and Günter Radons Phys. Rev. Lett. 102, 074102 – Published 18 February 2009

VANNITSEM, S., & NICOLIS, C. (1997). Lyapunov vectors and error growth patterns in a T21L3 quasigeostrophic model. Journal of the atmospheric sciences, 54(2), 347-361.

Schubert, S. and Lucarini, V. (2015), Covariant Lyapunov vectors of a quasi‐geostrophic baroclinic model: analysis of instabilities and feedbacks. Q.J.R. Meteorol. Soc., 141: 3040-3055. doi:10.1002/qj.2588

---

## Author Comment (AC1) · 10 Apr 2019

**Responses to Reviewers' Comments**

The author would like to thank both reviewers very much for their useful comments which helped improve the manuscript.

**Reviewer 1**

- Section 2

  Q:

  In the model, if I correctly understand, the "h" term is only a parameter (not a function). If so, I think the model is invariant under a zonal shift. This neutral transformation would imply a second vanishing Lyapunov exponent in the spectrum. Please, clarify this point.

  A:

  This is a misunderstanding; the model has no zonal symmetry. The topography $h = h(\lambda, \mu)$ is a function of space and is actually the real topography of the earth expanded into spherical harmonics. Also the diabatic source terms $S_i = S_i(\lambda, \mu)$ are functions of space, fitted from reanalysis data. So any symmetry is broken in the model and this is actually crucial for getting a realistic mean state and variability. This is clarified in the revised manuscript.

- Section 6.1

  Q:

  Concerning the last sentence in Sec. 6.1. I note that the equivalence between Lyapunov exponent fluctuations measured from Gram-Schmidt vectors and from covariant vectors, was detected already in Figs. 5 and 7 of Ref. [1].

  A:

  This is acknowledged in the revised manuscript.

  Q:

  In fact, the large fluctuations observed at the edges of the spectrum are not really surprising, at the light of the previous results on the diffusion coefficients in (Kuptsov and Politi, 2011) and [1].

  Q:

  This is referenced in the revised manuscript.

- Section 6.2

  Q:

  It is absolutely necessary to include one formula defining the fraction of explained variance, in order to ensure the self-consistency of the text.

  A:

  Done.

- Section 6.3

  Q:

  It is not said which is the total length of the time series used.

  A:

  The length of the time series is 25000 days; this information is now given in the revised manuscript.

Q:

The value of $\tau_r$ is "hidden" in Sec. 3.

A:

The value of $\tau_r$ is now again given in the results section.

Q:

The three methods used to measure $D_{j,j}$ are not fully clear to me. I think the author should make a list with the three methods specifying which formulas are used in each one. And which parameters are used. Now the explanation is hidden in the caption of Fig. 5, and is hardly understandable.
After reading it several times ...

A:

This part of the manuscript has been substantially revised. The different methods for estimating the rate function and the elements of the diffusion matrix are now explained more clearly and in more detail (see revised manuscript).

- Minor comments:

  - Q:
    Two lines after Eq. (13), I would write scalar product instead of norm, because this is what matters for Gram-Schmidt orthogonalisation. I suppose the energy norm is trivially related to the scalar product used.
    A:
    Corrected.

  - Q:
    Vectors should be always typed in bold face, also for Greek letters.
    A:
    Corrected.

  - Q:
    Orthogonality of the eigenvectors, Eq. (18), is better written after Eq. (16).
    A:
    Done.

  - Q:
    The equation in the text preceding Eq. (18) is apparently lacking of $-\lambda$.
    A:
    Corrected.

  - Q:
    When introducing Eq. (27), it would be important to cite at least (Touchette, 2009) again and to mention this is the Gärtner–Ellis theorem (if I'm not wrong).
    A:
    Done.

  - Q:
    The "log" symbol is missing in Eq. (35).
    A:
    Corrected.

  - Q:
    Figures 4–7 should be introduced in the text, one by one.

A:
Done.

– Q:
Last line of page 10. $\tau_c$ has not been defined.
A:
Corrected.

– Q:
Page 12, line 5. I don't appreciate smaller deviations of the rate function in this case than for the zero exponent.
A:
For $\tau = 1$ day and $\tau = 2$ days, there is some non-Gaussianity visible for the zero exponent but not for the 200th exponent as is shown in Figure 6 in the revised manuscript.

– Q:
In figures 5–8, I would use different colours for the lines in panel (b), at least for the coloured ones since they are not related to the same colours in panel (a).
A:
Done.

– Q:
In the conclusions, it is mentioned that the most unstable exponents exhibit slower convergence to the large-deviation limit. Let me to point out that this is fully consistent with [1].
A:
This is referenced in the revised manuscript.

– Q:
Labels (a), (b), etc. need to be included all the figures. This is critical in Fig. 9.
A:
Done.

**Reviewer 2**

- On section 1:
  Q:
  Line 8: based from what I understand: Could the covariance structure tell us something about how "close" or how "interactive" the various unstable and stable directions are? Could the covariance structure be related to the investigations of the inertial manifold using the angles between Lyapunov vectors. Maybe the work along the lines of Yang et al (2009) should be referenced here as a motivation.
  A:
  This is an interesting point but would clearly need the use of the covariant Lyapunov vectors. It is mentioned in the revised manuscript as a possible future research line and the work by Yang et al (2009) is referenced.

- On Section 2:
  Q:
  I think a short concise table listing all parameters of the model with their dimensional

and a dimensional values would be beneficial to introduce the model setup.
A:
Tables listing all the variables and parameters of the model have been included in the revised manuscript.

- On Section 3:
  Q:
  It should be noted that the mean of the finite time Lyapunov exponents are in fact average growth rates of linear perturbations of the system. But the finite time LEs are not directly the growth rate of those perturbation. In fact one can define backward, forward and covariant LEs. There is a good review paper on this by Kuptsov and Parlitz which explains this distinction. I think using the FTLEs of the Gram Schmidt algorithm is alright, but it should be better clarified what type of FTLEs they actually are.
  A:
  Some comments on this have been added to the manuscript and the paper by Kuptsov and Parlitz is cited. In the limit of large integration time $\tau$, which is the focus of the present paper, the three types of FTLEs are actually equivalent and the backward FTLEs are easiest to calculate.

- On Section 4:
  Q:
  I think this section should motivate better why one should use EOFs and what would be potentially alternatives to this approach.
  A:
  A couple of comments have been added here (see revised manuscript).

- On Section 6:

  - Section 6.1:
    Q:
    Since the model is zonally symmetric there should be two zero exponents. Can you verify this and could you include this in this discussion?
    A:
    This is a misunderstanding; the model has no zonal symmetry. The topography $h = h(\lambda, \mu)$ is a function of space and is actually the real topography of the earth expanded into spherical harmonics. Also the diabatic source terms $S_i = S_i(\lambda, \mu)$ are functions of space, fitted from reanalysis data. So any symmetry is broken in the model and this is actually crucial for getting a realistic mean state and variability. This is clarified in the revised manuscript.

    Q:
    Figure 2: This result should be referenced with the findings about fluctuations of the LE for covariant, backward and forward exponents in Vannitsem, Lucarini (2016). I think when you study collective excitations this is an interesting different viewpoint.
    A:
    Done.

  - Section 6.2:
    Q:
    I think it would be helpful to present the matrix D as well as a surface plot and also

use the first EOF and second EOF to see what parts of the D matrix are actually reconstructed using the EOF method.

A:

I tried this but it turned out not to be really useful as the matrix D is strongly dominated by the diagonal and the leading EOFs do not explain that much variance, even in the diffusion limit. I have added a plot showing the correlation of selected FTLEs with all the other exponents (see Figure 5 in the revised manuscript).

- Section 7:

Q:

That no clear time scale separation is found is probably because QG equations are scale filtered quations. Similar results were found before in QG models (Vannitsem 1997, Schubert 2015, Vannitsem 2016).

A:

I agree and a comment along this line has been added.

Q:

Can the collective excitations be traced to any anomalous behaviour in the nonlinear background state x? I think that would an interesting addition but of course not necessary in order to do this study.

A:

A separate study is actually underway by the author linking FTLEs to underlying weather regimes in the model state.

---

## Referee Report (RR1)

The manuscript has been improved with respect to the previous version. The results are interesting and new. However, the presentation is still messy at several points. In my opinion, the manuscript could be publishable after substantial changes. I provide the details below.

**A.  Large deviation theory**

The presentation of the large deviation theory is a mess, specially considering the presentation of the block averaging method. Is not enough to say that for large enough $\tau$ FTLEs are virtually uncorrelated?

I think it is much simpler to write a large deviation formula for the $j$-th exponent

$$P(\Lambda_j^{(\tau)} = z) \sim e^{-\tau I_j(z)}$$

with rate function

$$I_j(z) = \sup_\theta [\theta z - \gamma_j(\theta)]$$

and

$$\gamma_j(\theta) = \lim_{\tau \to \infty} \frac{1}{\tau} \ln \left\langle e^{\tau \theta \Lambda_j^{(\tau)}} \right\rangle$$

That's all. Equations (35) and (36) in the text are recovered defining $\theta' = \tau\theta$, but they are much more difficult to understand. After all the calculations in the manuscript, Eq. (36) presents a rate function that apparently depends on $\tau$; while it is claimed it doesn't in the previous sentence (!). (This also applies to Eq. (46).)

**B.  Inertial manifold**

At the end of Sec. 6.2 the new paragraph on the inertial manifold is not correct. It points to a dimensionality of the inertial manifold ($\approx 125$) much smaller than the dimensionality of the attractor obtained by the Kaplan-Yorke formula. This is not possible: the dimension of the attractor is necessarily smaller than the dimension of the inertial manifold.

**C.  Correlation length**

1. The formula for the correlation length in Eq. (57) is puzzling. Where it comes from? what hypotheses have been assumed?

2. The text suggests that the crossover $\tau$ for convergence to the large-deviation regime is related to the correlation time of the FTLE. I disagree with this interpretation for two reasons. First, the correlation lengths observed are not much different in comparison with the crossover $\tau$'s. Second, in spatially extended systems it is typical that correlation times are insensitive to the system size, while crossover $\tau$ scales with the system size as $L^z$, $z > 0$. This indicates a different operate in each case (Pazó et al, 2013).

**D.  Legendre transform method**

I encourage the author to emphasize in the conclusions that the Legendre transform approach appears to be superior to the probability density approach in order to find the rate function (assumed the conditions of the Gärtner-Ellis theorem hold). Moreover the Legendre transform method yields diffusion coefficients fully consistent their direct measurement from the data.

**E.  Figures**

1. In Figs. 2(c) and 2(d) the factor $\tau^{1/2}$ should appear in the y-axis label.

2. I have a suggestion, which can be followed or not: In Figs. 3 (a) and 3(b), wouldn't it be clearer to set log scale for the y-axis?

3. Figure 4 is more clear writing $y_j^{(\tau)}$ in the y-axis label.

4. Contrary to what is claimed in the author's response, the labels '(a), '(b)', etc. are not included in the revised ms.

**F.  Typos**

1. Page 17, line 8, $n \to L/n$.

2. Page 21, line 4, to $\to$ too.

---

## Author Response (AR2)

[revised manuscript text omitted]

A:

The references have been removed.

---

## Author Response (AR3)

**Responses to Reviewer's and Editor's Comments**

I recommend you to:

1) write $y_1$, $y_2$, and $y_3$ explicitly in the $Y$-axis, instead of using an ambiguous term as "amplitude",
It is not the temporal amplitude, $y_j$, but the components of the patterns themselves, $e_{j,i}^{(\tau)}$. This is now clarified and linked to the text.

2) mention in the figure caption Eq.(21), so that the reader could easily link the plot with the quantities defined in the text,
Eq.(20) is referred to in the caption of Figure 4.

3) in Fig. 4 also replace $j$ in the $X$-axis by $i$ to be consistent with the notation of the paper.
Done.